# Monitoring of Non-Lame Horses and Horses with Unilateral Hindlimb Lameness at Rest with the Aid of Accelerometers

**DOI:** 10.3390/s24227203

**Published:** 2024-11-11

**Authors:** Anja Uellendahl, Johannes P. Schramel, Alexander Tichy, Christian Peham

**Affiliations:** 1Movement Science Group, University Equine Hospital, University of Veterinary Medicine, 1210 Vienna, Austria; anja.uellendahl@web.de (A.U.); johannes.schramel@vetmeduni.ac.at (J.P.S.); 2Platform for Bioinformatics and Biostatistics, Centre of Biological Sciences, University of Veterinary Medicine, 1210 Vienna, Austria; alexander.tichy@vetmeduni.ac.at

**Keywords:** resting pattern, horse, unilateral hindlimb lameness, accelerometer, comparative study

## Abstract

The aim of this study was to determine whether horses exhibiting unilateral hindlimb lameness unload (rest) the lame limb more than the contralateral limb. The resting/unloading of the hindlimbs and the time spent lying down were measured using accelerometers. Ten non-lame horses and 20 lame horses were recruited for participation and monitored for 11 h overnight with accelerometers (MSR145, sampling rate: 1 Hz, and measuring range: ±15 g) attached to the lateral metatarsal and metacarpal regions of each limb. Metatarsal and metacarpal orientation were used to determine whether the limb was unloaded (rested) or loaded, respectively, or whether the horses were lying down. The relation of resting time between non-lame and lame limbs (non-lame/lame: 0.85 ± 1.2) of the lame horses differed significantly (*p* = 0.035) from that of the non-lame horses (right/left: 1.08 ± 0.47). Non-lame horses rested their hindlimbs evenly (left: 15 ± 10%; right: 17 ± 16%). Horses with unilateral hindlimb lameness unloaded the lame limb longer (lame limb: 61.8 ± 25.3%, non-lame limb: 38.2 ± 25.3%) than their contralateral limb. The lame horses (13 ± 11%) lay down longer (*p* = 0.012) than the non-lame horses (3 ± 6%). The degree of lameness determined by the participating veterinarians (Vet Score) (r = −0.691, *p* < 0.01) and the asymmetry evaluated by the lameness locator (ALL) (r = −0.426, *p* = 0.019) correlated with the resting ratio (rest time ratio). Both factors were also correlated with the time spent lying down (Vet Score (r = 0.364, *p* = 0.048) and the ALL (r = 0.398, *p* = 0.03)). The ALL and VET Score were significantly correlated (r = 0.557, *p* = 0.01). The results of this study provide a good baseline for future research into how individual resting patterns may help to detect pain.

## 1. Introduction

Lameness in horses is a widespread problem that significantly affects their well-being and performance. In the past, lameness has been diagnosed by subjective assessments of a veterinarian, but currently, sensor-based objective methods are being more and more used for this purpose [1]. Sensors such as accelerometers provide reliable movement analysis and enable precise detection of lameness [2]. Perception, even by experienced veterinarians, is often distorted by external influences [3]. The accuracy of human lameness detection depends on the severity of lameness [4]. Mild lameness can be better detected by assistive technology [5,6]. Early detection of lameness, in general, is relevant for animal welfare and is therefore essential for prompt diagnosis and treatment. This reduces a horse’s rest time and minimizes the possibility of aggravation of mild lameness [7].

Based on a pilot study [8], this study compared the resting patterns of non-lame and lame horses. Non-lame horses usually display a wide range of behaviors, including exercise, feeding, and social interactions. They tend to move more freely and engage in regular activities without signs of discomfort. In contrast, lame horses commonly show increased signs of discomfort and pain, such as altered movement patterns, reduced activity, and postural changes [9]. These horses may also display patterns such as limb unloading, head nodding, and reluctance to move. In addition, lame horses spend more time lying down and show more restless behavior than non-lame horses [9]. These differences may be attributed to the pain and discomfort associated with lameness and therefore provide valuable insights into a horse’s well-being [10]. These behavioral changes have already been studied in dairy cows and, in some cases, in pregnant sows housed in groups [11,12,13,14,15].

The aim of this study was to assess limb loading and unloading times alongside the time spent lying down in two groups of horses (non-lame and those exhibiting unilateral hindlimb lameness) utilizing accelerometers, which have previously been documented to be successful for postural measurements in horses [8]. This study aims to build on previous research on other species [15] and gain a baseline of information regarding resting/unloading patterns in horses exhibiting lameness.

The following hypotheses were tested in this study:

**Hypothesis** **1.**
*The lame limb is loaded for a significantly shorter period of time than the non-lame limb. This can be defined by (a) resting (unloading) a single hindlimb and (b) the time spent lying down.*


**Hypothesis** **2.**
*Non-lame horses load and unload the limbs evenly.*


**Hypothesis** **3.**
*Lame horses lie down for longer periods of time than non-lame horses.*


## 2. Materials and Methods

### 2.1. Ethical Approval

This study was reviewed and endorsed by the Ministry for Energy Transition, Agriculture, Environment, Nature and Digitalisation in Kiel, Germany and the Ethics and Animal Welfare Committee of the University of Veterinary Medicine, Vienna. The signs created for this purpose are V 242-9770/2022 and ETK-08/06/2017.

### 2.2. Animals and Experimental Design

A group of 30 horses were recruited for participation. These horses were all inpatients in an equine hospital and were selected from the hospital’s caseload. All participating horses were patients that presented to the equine hospital for veterinary investigations and treatment and had been inpatients at the hospital for one to two days prior to participation in this study. The data were collected over one year. In addition, horses examined during the pilot study that fulfilled the inclusion criteria were included. The selected horses were of different breeds, ages, and sizes (Table 1). The participating horses were observed in their stables overnight during a measurement period of approximately 11 h. Out of the participating horses, ten horses were not lame on the day of examination, had no recent history of lameness, and had been ridden regularly. Horses were classified as non-lame based on the absence of lameness on the day of examination.

Twenty horses had unilateral hindlimb lameness, which was confirmed by the Equinosis^®^ Lameness Locator (ALL) (104 E Broadway, Columbia, MO, USA) (https://equinosis.com/pferd-lameness-messung-pferde-lahmheitserkennung-equine-lameness-locator/, accessed on 31 August 2024), by a well-experienced veterinarian, and by a veterinarian with less experience (Vet Score); 12 of the horses were right hind (RH) lame and 8 were left hind (LH) lame. Diagnostic regional anesthesia was utilized as part of the lameness investigations to confirm that the lameness exhibited was unilateral. Lameness was graded using the American Association of Equine Practitioners (AAEP)’s Lameness Scale [16,17] (Appendix A), where grade 1 denotes mild, inconsistent lameness that is difficult to observe and grade 4 denotes lameness that is obvious at walk. Half units are generally used when there is more than one evaluating veterinarian [1] and were thus used in this study because two veterinarians assessed lameness. An average score was obtained if they were different between the veterinarians.

All horses were assessed at walk and trot in a straight line on firm ground. In addition, they were lunged at walk, trot, and canter on a soft surface if their lameness allowed it (canter was not always possible for those with grade 4 lameness).

The degree of lameness for all horses was always in the range of 1–4 out of 5 (Table 1) using the AAEP lameness grading scale.

Each horse was monitored in a stable (approximately 4 × 4 m) overnight for 11 h without interruption. The box was bedded with a mixture of wooden chips and straw. During this time, the horses were not disturbed by mucking out, being groomed, or being taken out of the box. All measurements were taken at night in order to minimize the external stimuli, which are normal for a daily routine in an equine clinic.

The accelerometers (MSR145, sampling rate: 1 Hz, and measuring range: ±15 g) were fastened to the limbs with an elasticated Velcro strap firmly attached to the accelerometers. A thin layer of cohesive bandage was placed on the mid-metatarsal and metacarpal region of each limb, over which the accelerometers were attached perpendicular to the ground (Figure 1). An additional layer of cohesive bandage was applied over the sensor to protect it.

Following the application of the sensors, all horses were monitored until every measurable activity except lying down was observed once to ensure that the sensors were tolerated.

Each horse had a conspecific in an adjacent box or across the aisle.

### 2.3. Accelerometers

Four identical accelerometers (MSR145, sampling rate: 1 Hz, and measuring range: ±15 g, https://www.msr.ch/de/produkt/datenlogger-temperatur-feuchte-druck-beschleunigung-msr145/, accessed on 2 August 2024) were used for this measurement technique. The dimensions were 27 × 63 × 53 mm (width × height × length) with a weight of approximately 20 g. The 230 mAh battery ensured measurements over several consecutive days. A built-in LED display signaled whether or not the device was recording, a malfunction, or a low battery status through different colors and flashing. The data loggers were charged using a USB cable, and the data were extracted using the manufacturer’s software (MSR software, version: MSR 5.32.02). The data rate/sampling rate was 1 Hz, and the measuring range was ±15 g. The corresponding datasheet can be seen in the appendix (Appendix B: Data Sheet Mini Data Logger MSR145).

MSR145 is a three-axis accelerometer, and several of these were used in this study. Each accelerometer was assigned to a specific limb for easier orientation and data tracking (static situation). With identical fixation on each limb, the alignment could be made in relation to a horse’s limb and the ground. This study aimed to observe the vertical acceleration of the limbs to determine whether a horse was changing its stance or resting one limb.

To detect motion events, the accelerometers detected changes in gravity to measure vertical acceleration. The vector of the degree of vertical acceleration due to gravity at stance was always 1 g (=9.81 m/s²). The z-axis was aligned vertically. Any change was always measured at the vertical position as it had the greatest deflection and the highest sensitivity. In addition to the vertical acceleration due to gravity, the change in the angle of the sensors in space could be determined (Appendix C: Angle calculation as a second method).

### 2.4. Data Handling

A mean value was calculated for each case. The orientation was calculated with MS-Excel 2013 (Microsoft cooperation^®^), while statistical analyses and the generation of graphical data illustrations were carried out using SPSS 29.0 (IBM^®^). The latter was also used to create boxplots and other graphical representations.

The accelerometers were able to distinguish between limb unloading and the horse lying down; the duration of these events was logged (in minutes).

In addition, a mean value for the percentage ratio of the resting time was created for the non-lame comparison group in order to be able to make a summary statement about the resting patterns of the non-lame horses. The total time of each individual activity was also set in relation to the total measuring time.

Resting relation is defined as non-lame resting time/lame resting time within the lame group of horses and right limb resting time/left limb resting time within the non-lame group of horses. The data of both hindlimbs were analyzed. Asymmetry represents the minimal and maximal difference of the Equinosis Lameness Locator^®^ in millimeters.

### 2.5. Statistics

A Kolmogorov–Smirnov test was used to determine whether the data were normally distributed. Since the data were not normally distributed, a Mann–Whitney test was used to compare the groups. In addition, the Spearman–Rho correlation coefficient between the lameness score, the asymmetry evaluated by the lameness locator, and the relation of the resting pattern were calculated.

In all statistical analyses, a *p*-value less than 0.05 (*p* < 0.05) was interpreted as significant.

## 3. Results

The evaluation of the datasets resulted in the following values for the resting patterns “resting one single hindlimb” and “lying down”. The number of each horse and the percentage distribution are summarized in Table 2 and Table 3:

For the percentage distribution between the lame limb and the non-lame limb within the lame horses, see Appendix C: Percentage distribution of non-lame and left or right hindlimb lame horses resting their hindlimbs.

The resting relation for the lame horses was statistically significant (*p* = 0.035). The lame horses (13 ± 11%) lay down longer (*p* = 0.01) than the non-lame horses (3 ± 6%) (Figure 2).

Non-lame horses rested their hindlimbs evenly (left: 15 ± 10%; right: 17 ± 16%). Lame horses spent more time lying down than non-lame horses (*p* = 0.025) (Figure 3).

The asymmetry evaluated by the lameness locator (ALL) showed a difference (*p* = 0.025) between the lame group (10.5 ± 5.6 mm) and the non-lame group (5.8 ± 3.4 mm) of horses. The resting ratio (rest time ratio, in minutes) of the lame horses (0.86 ± 1.2) was lower (*p* = 0.035) than that of the non-lame horses (1.08 ± 0.47), while the time spent lying down was higher (*p* = 0.013) in the lame horses (84 ± 71) compared to the non-lame horses (16 ± 35). The Vet Score (r = −0.691, *p* < 0.01) and the ALL (r = −0.426, *p* = 0.019) were correlated with the resting ratio (rest time ratio). Both factors were also correlated with the time spent lying down (Vet Score (r = 0.364, *p* = 0.048) and ALL (r = 0.398, *p* = 0.03). The ALL and Vet Score were correlated (r = 0.557, *p* = 0.01). The mean asymmetry evaluated by the lameness locator (ALL) was different (*p* = 0.025) in lame horses (10.5 ± 5.6 mm) compared to non-lame horses (5.8 ± 3.4 mm) (Figure 4).

Although the results are significant, on average, individual non-lame horses showed a similar resting pattern to horses with unilateral hindlimb lameness.

## 4. Discussion

In accordance with our hypotheses, non-lame horses rested their hindlimbs evenly, and in the case of unilateral hind lameness, the lame limb was unloaded more frequently than the non-lame limb.

However, if the results of individual horses are analyzed before the data are averaged, clear differences can be observed. Although the results are significant, on average, individual non-lame horses showed a similar resting pattern to horses with unilateral and usually less severe hindlimb lameness. Due to the lameness severity in the participating horses, the results were not necessarily transferable to horses that only showed hindlimb lameness when ridden and that did not show visible or measurable asymmetry outside the reference range. Horses might become used to pain and therefore show a less severe or lower grade of lameness. However, this had no impact on this study because we only considered whether they were lame or not on the day of the examination.

It would be advantageous if the individual resting behavior were known since horses prefer one side (handedness) [18,19,20], and it is also unclear to what extent handedness might have influenced our findings. Laterality is evident in foals [21], which can affect their training [22], well-being [23], or physical abilities [24] later on. Handedness is therefore composed of three described asymmetries. To demonstrate the influence or distortion of handedness, it is necessary to conduct a study using a larger sample size and by taking into account the handedness/laterality of the horses [25].

Lameness is dynamic and can change over time [26]. Therefore, it is important to analyze enough movement cycles in order to obtain a stable result [27,28]. With long-term lameness, horses can overload the contralateral hind limb and cause discomfort [10], which can also affect resting behavior.

The differences in pain tolerance among the various breeds included in this study may have influenced the results. For instance, Thoroughbreds, known for their sensitivity, might show more obvious pain behaviors, while Warmbloods, which are typically more stoic, might display subtler signs, leading to an underestimation of their pain [29]. Ponies like Shetlands might show fewer obvious signs due to a higher pain tolerance. Future studies should focus on a single breed to minimize this variance. Pain assessment tools, like ethograms, should be adapted for different breeds to ensure accurate detection. This study also found that horses with unilateral hindlimb lameness lay down more often and for longer periods, supporting the hypothesis that lame horses rest more due to pain. This correlation has also been shown in dairy cows [11,12,13].

Asymmetry was found to be correlated with rest time ratio. Both factors were also correlated with lying time and the ALL. This is consistent with the findings of Blackie et al. [11], who found a correlation between lameness severity and rest time in cattle. The horses were included/separated based on the Vet Score. A perfect match between the Vet Score and the ALL is very unlikely. However, the statistical comparison showed a significant correlation between the Vet Score and the ALL in this study.

However, the results were influenced by factors such as the duration, cause, and type of lameness as well as the individual pain tolerance of the horse. Numerous studies on the detection and treatment of pain [30,31,32] have shown this.

Horse 6 demonstrates the complexity of this issue as its resting behavior was the opposite of that of other horses. One reason for this could be an error in the lameness evaluation. However, this might also be an example of an atypical lameness pattern. Therefore, further research in this area is needed [33] to provide clear guidelines for veterinarians.

In this study, the cause of the horses’ lameness was not examined. Although different diseases result in different levels of pain, this was assumed to be reflected in the degree of lameness; so, the etiology was not considered when assessing resting patterns. The Equinosis Lameness Locator^®^ was used to objectively detect lameness, including the presence of subtle signs that might be overlooked otherwise [34]. This may explain the differences between the Vet Score and the Equinosis Lameness Locator^®^. All horses were examined by the same veterinarians, and the Equinosis Lameness Locator^®^ confirmed their assessments [6,35,36].

An observation period of 11 h provided sufficient data to detect resting patterns, although future studies could be conducted to collect data over shorter or longer periods of time. Shorter periods of examination could be beneficial for horse owners if the findings prove to be reliable. Comparing the resting patterns of lame and non-lame horses could also help to detect behavioral indicators of lameness beyond those visible in pain ethograms [37,38]. Measurements for this study were taken at night to capture natural resting patterns with minimal disturbance, although the clinical setting might have influenced the results due to the unfamiliar surroundings, such as in the case of the non-lame horses that did not lie down. Uncertainty in the data can certainly be reduced if the horses are analyzed in their usual environment [39,40].

The accelerometers effectively detected resting patterns [41] by measuring small variations in the x-, y-, and z-axis values, reflecting different types of limb unloading. This is in accordance with many studies that have analyzed behavioral patterns [14,42,43,44,45,46]. The advantages of accelerometers are their continuous monitoring capacity and ability to detect subtle changes at rest and during exercise. These objective data could be used to track the progress of treatment [47]. Some horses rested their hindlimbs differently, with more angulation in the fetlock joint or by placing the lame limb forward [10,47]. However, “weight shifting”, which is a pain indicator, could not be measured by the accelerometers, which rely on changes in limb angle. Even overhead cameras could not reliably detect weight shifts; they could be observed more accurately by the human eye. Monitoring how horses get up and lie down, which would require cameras, could be another useful approach but was not performed in this study. Using accelerometers on a daily basis would be beneficial but might be impractical for owners. However, observing resting patterns during the day, rather than only at night, could be useful.

Horses typically sleep in short bursts during the day and night, accumulating 2–3 h of deep sleep (SWS) and 30–60 min of REM sleep in a 24 h period [47,48]. Sleep patterns vary depending on age, environment, and health, with younger horses sleeping more and those in comfortable environments being more likely to gain REM sleep [49]. Chronic pain, for example, from lameness, disrupts sleep by reducing lying time and limiting REM sleep. Horses with chronic orthopedic problems change their resting pattern, which causes discomfort and reduces restful sleep. This behavior is consistent with our findings.

## 5. Conclusions

To summarize, this study showed that accelerometers offer a good way to monitor resting patterns in lame and non-lame horses and provided a potential baseline for the development of possible monitoring systems.

## Figures and Tables

**Figure 1 sensors-24-07203-f001:**
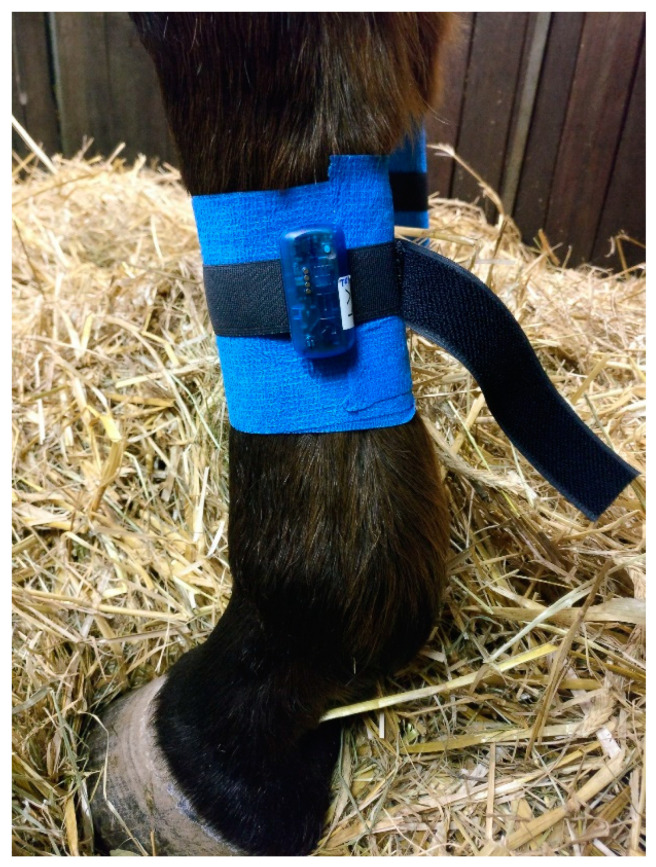
The method of attaching the accelerometers (MSR145, sampling rate: 1 Hz, and measuring range: ±15 g) to the equine extremity laterally in the mid-region of the metacarpal and metatarsal bone with a Velcro fastener and one layer of elastic band.

**Figure 2 sensors-24-07203-f002:**
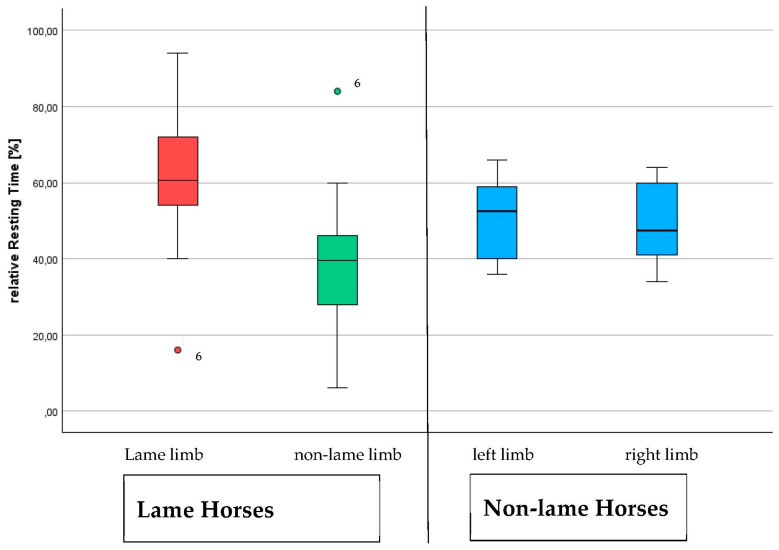
Comparison of the percentage distribution of the hindlimb resting time between limbs for lame and non-lame horses participating in this study to measure loading and unloading patterns of hindlimbs in lame horses compared to non-lame horses. Horse 6 is marked as an outlier in the first two boxplots (shown by the circles and number 6).

**Figure 3 sensors-24-07203-f003:**
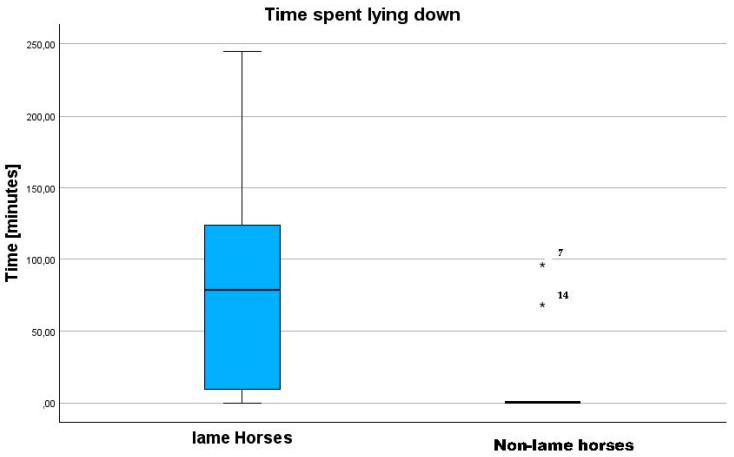
Comparison and percentage distribution of the time spent lying down between the lame horses and the non-lame horses participating in this study to measure loading and unloading patterns in lame horses and non-lame horses. Horses 7 and 14 are marked as outliers in the right boxplot (shown by asterisks and the numbers 7 and 14).

**Figure 4 sensors-24-07203-f004:**
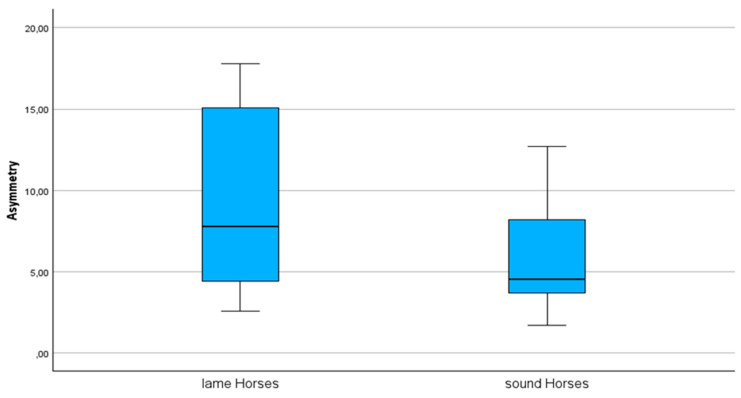
Objectively measured asymmetry between 20 horses with unilateral hindlimb lameness and 10 non-lame horses participating in this study to evaluate loading and unloading patterns in lame limbs compared to non-lame limbs (measured in millimeters (mm)).

**Table 1 sensors-24-07203-t001:** Details of 20 horses (breed, year of birth, sex, group, and lameness degree using the AAEP scale) with unilateral hindlimb lameness and 10 non-lame horses that underwent evaluation of resting patterns wearing hindlimb accelerometers for 11 h.

Horse Number	Breed	Year of Birth	Sex	Non-Lame	Lameness (Side, Degree)	Test Date
1	Holstein	2012	Female (F)		RH (right hind) (3/5)	30 January 2020
2	Pura Raza Espanola	2003	Male (M)		RH (3/5)	3 February 2020
3	Holstein	2003	M	X		2 March 2020
4	Holstein	2012	M		RH (3/5)	9 March 2020
5	Hanoverian	2011	M		LH (left hind) (3/5)	22 March 2020
6	Hanoverian	2013	F		LH (3/5)	30 March 2020
7	Holstein	2011	M	X		31 March 2020
8	Oldenburg	2011	F		LH (2.5/5)	19 April 2020
9	Hanoverian	2013	F		LH (1.5/5)	27 April 2020
10	German Sporthorse	2009	F		RH (3.5/5)	19 May 2020
11	Holstein	2006	F		RH (2/5)	5 June 2020
12	Pinto Hunter	2001	F		LH (3.5/5)	10 June 2020
13	Pony	2003	F		LH (4/5)	21 July 2020
14	Icelandic horse	2007	M	X		31 July 2020
15	Hanoverian	2014	M		LH (1.5/5)	20 August 2020
16	Hanoverian	1993	M		RH (3/5)	26 August 2020
17	Hanoverian	2010	M		LH (3.5/5)	21 October 2020
18	Trakehner	2017	M	X		15 October 2020
19	Hanoverian	2008	M	X		19 October 2020
20	Holstein	2005	M	X		20 October 2020
21	Hanoverian	2014	M		RH (2/5)	20 October 2020
22	Hanoverian	2010	M		RH (3.5/5)	17 November 2020
23	Hanoverian	2013	M		RH (3/5)	17 November 2020
24	Hanoverian	2012	M		RH (3.5/5)	18 November 2020
25	Haflinger	2010	M	X		1 August 2017
26	Haflinger	2011	M	X		2 August 2017
27	Standardbred	2002	M	X		19 August 2017
28	Czech Warmblood	1992	F		RH (3.5/5)	21 July 2017
29	Norwegian Cross	2005	F	X		17 August 2017
30	Standardbred	1992	M		RH (4/5)	4 August 2017

**Table 2 sensors-24-07203-t002:** Summary of lame horses, including lameness as determined by the veterinarians (Vet Score), asymmetry (defined as the minimal and maximal difference evaluated by the Equinosis Lameness Locator^®^ in millimeter), the resting time between the lame limb and the non-lame limb, the time spent lying down in minutes, and the resting relation (defined as non-lame resting time/lame resting time) between the rested limbs for horses participating in a study evaluating loading and unloading times of non-lame limbs compared to lame limbs.

		Vet Score (AAEP Scale)	Lameness Locator (mm)	Rest/Lame (%)	Rest/Non-Lame (%)	Time Spent Lying Down (Minutes)	Resting Relation
Horse 1	lame	3	15.1	86	14	184	0.16
Horse 2	lame	3	7.4	66	34	101	0.52
Horse 4	lame	3	17	54	46	245	0.85
Horse 5	lame	3	2.6	58	42	19	0.72
Horse 6	lame	3	17.8	16	84	134	5.25
Horse 8	lame	2.5	8.2	57	43	0	0.75
Horse 9	lame	1.5	4.4	40	60	72	1.50
Horse 10	lame	3.5	4.3	72	28	114	0.39
Horse 11	lame	2	7.2	63	37	56	0.59
Horse 12	lame	3.5	12.4	94	6	106	0.06
Horse 13	lame	4	18.3	100	0	143	0.00
Horse 15	lame	1.5	4.7	29	71	84	2.45
Horse 16	lame	3	6.5	47	53	0	1.13
Horse 17	lame	3.5	11.9	100	0	0	0.00
Horse 21	lame	2	7.7	47	53	114	1.13
Horse 22	lame	3.5	7.7	79	21	51	0.27
Horse 23	lame	3	7.9	66	34	73	0.52
Horse 24	lame	3.5	22	73	27	184	0.37
Horse 28	lame	3.5	9.9	84	16	0	0.19
Horse 30	lame	4	17.2	93	7	0	0.08

**Table 3 sensors-24-07203-t003:** Summary of non-lame horses, including lameness as determined by the veterinarians (Vet Score), asymmetry (defined as the minimal and maximal difference evaluated by the Equinosis Lameness Locator® in millimeter), the resting time between the left limb and the right limb, the time spent lying down and the resting relation between the rested limbs (defined as right limb resting time/left limb resting time) for horses participating in a study evaluating loading and unloading times of right limbs compared to left limbs.

		Vet Score (AAEP)	Lameness Locator (mm)	Rest/Left (%)	Rest/Right (%)	Time Spent Lying Down (Minutes)	Resting Relation
Horse 3	non-lame	0	4.7	36	64	0	1.78
Horse 7	non-lame	0	9.5	41	59	96	1.44
Horse 14	non-lame	0	3	59	41	68	0.69
Horse 18	non-lame	0	1.7	55	45	0	0.82
Horse 19	non-lame	0	4.3	37	63	1	1.70
Horse 20	non-lame	0	3.7	40	60	0	1.50
Horse 25	non-lame	0	12.7	63	37	1	0.59
Horse 26	non-lame	0	4.4	66	34	0	0.52
Horse 27	non-lame	0	5.9	52	48	0	0.92
Horse 29	non-lame	0	8.2	53	47	0	0.89

## Data Availability

Data are available at the University of Veterinary Medicine and can be downloaded upon request.

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
