# Peer review of "Monitoring of Non-Lame Horses and Horses with Unilateral Hindlimb Lameness at Rest with the Aid of Accelerometers"

_sensors, 2024, doi:10.3390/s24227203_

Round 1
Reviewer 1 Report
Comments and Suggestions for Authors
Referee’s report Sensors
This could be a potentially interesting paper - but I think that the sample sizes are too small & the SDs too large to draw meaningful conclusions. There was no repeatability of the observations, to determine how consistently they occurred.
There is inadequate description of the methods to be able to reproduce the study. There is overinterpretation of the results with potentially misleading conclusions.
The authors stated that 'This work was carried out to investigate the early detection of lameness', however the design of the study does not permit this to be performed.
The early recognition of lameness / musculoskeletal pain by a caregiver stimulating the request for veterinary investigation would certainly be of benefit. We know that riders and trainers are poor at recognising lameness based on numerous studies.
Some of the text reflects lack of adequate knowledge about the literature relating to equine lameness manifestations, behavioural adaptations to pain and sleep.
Many sentences could be constructed in a more succinct manner to both reduce the word count & to make the meaning clearer. The English requires substantial improvement.
In my opinion very substantial revision of the manuscript is required for a study which could only be regarded as a pilot study from which no definitive conclusions can be drawn.
Title
‘and horses with unilateral hindlimb lameness’ would be a preferably wording
‘at stance’ is misleading because you were evaluating their behaviour in a stable including lying down
Abstract
Line 13 monitored not detected
Line 14 please use correct anatomical nomenclature - not cannon bone
metatarsal region if a hindlimb
It should be made clear that both hindlimbs were instrumented (here & in the M&M)
Line 16 lying down
rest patterns
Line 18
‘Horses with unilateral hind lameness rest the affected leg significantly longer’
This will surely depend on the severity of the lameness
What was the statistical power of the study?
Line 20 If the result was expressed to 2 decimal places - which is appropriate - the result is not statistically significant (p=0.05).
I suspect that your study sample is too small to draw meaningful results, & there is no repeatability over several days of observation. This cannot be considered anything other than a pilot study.
‘11 hours was considered to be sufficient’ On what bases could this statement be made – you did not compare variable periods of assessment.
Was there any human presence during this time?
Was video surveillance also used? - & if not why not?
What does ' the total of the lame horses' mean?
Line 21 lay down for significantly longer
This will depend on degree of discomfort - some lame horses have reduced lying down & experience sleep deprivation.
Introduction
Line 28 horses' behaviour
Accelerometers only tell you a limited amount about behaviour - when & to want extent a horse is moving.
Line 28 its creation - what does its refer to?
Line 58 what does 'in this study ' refer to?
Line 61 You need to look at Torcivia & McDonnell's work
Line 81 lay down
Materials and Methods
Line 88 Were these horses in their home environments or at a clinic? If at a clinic for how long had they been there before acquisition of the results? This information should be in the M&M.
How may these factors have influenced the results?
Line 89 what is meany by 'the revised criteria'?
Line 92 Determined how? Under what circumstances were the non-lame horses examined. Were these horses in regular work? how can you say 'no orthopaedic disease'? They could have had clinically silent radiological abnormalities consistent with osteoarthritis.
Non-lame on day of examination and in regular work, working willingly?
Line 94 Under what circumstances were the horses evaluated to determine they were grade 1/5 on the AAEP scale?
Line 97 How can you say with certainty that a horse is unilaterally lame unless the horse is blocked to abolish the lameness and the horse does not switch lameness to the contralateral limb?
Table 1 All tables & figures should have comprehensive legends so that these can be read independently from the text
How were the horses selected?
There are a high proportion of horses with grade 4/5 lameness - that is severe and atypical of the sports horse.
Please check all English spellings of the breeds and define any abbreviations e.g., PRE
Line 108 metacarpal or metatarsal region? Not cannon bone
Which limb(s)??
Line 116 why not lying down?
Where was the person - what was their experience - what did they record - over what time period were horses observed?
Line 118
‘Each time, another horse was in the same stable’
What does this mean?
Line 121 So 2 per horse, one on each hindlimb ?
Or (line 134) So were 4 accelerometers used per horse, one on each limb?
Line 143 Avoid repetition!
Line 150 ‘this had the greatest deflection and the highest sensitivity’ - based on this work or previous work??
Not clear!
Line 155 within not among
Line 167 'certain ratios of the values were exceeded ' - what does this mean?
So which horses were excluded
How many lame & non-lame horses?
Any why do the tables include these excluded horses?
How were the non-lame horses determined to be non-lame? Under what circumstances were they evaluated?
What was the duration of lameness in the lame horses?
Line 172 Why was log transformation performed?
Results
Table 2 All tables & figures should have comprehensive legends so that these can be read independently from the text
n=9
N=10 in original definition
3 & 26 outliers?
Table 3 Why are horses 6 & 15 outliers?
Large SD
n=8
6 grade 3/5
15 grade 1-2/5
How can a horse be between grades?
Table 4
16 & 21 outliers
large SD
Why are there 13 horses when you said in the text that there were 12 horses with RH lameness?
In table 1 horse 7 is listed as non-lame
16 grade 3/5
21 grade 2/5
Line 189 Why different for 3 & 26?
Line 196 'The total of the lame horses (13±11%) 196 lay significantly longer than sound horses (3±6%). '
What does this mean - that only 3 non-lame horses lay down at all?
Line 200 sound group, healthy group - please be consistent in terminology
Non-lame may be better
The Results need to be expressed more succinctly
Please express percentage results to no more than 1 decimal place
Line 202 What does 'settle at a similar time in %' actually mean?
This describes, as an example, how the text could be shorter.
The lame hind limb was rested for longer (61.8% ±25.3%) than the non-lame limb (38.2%±25.3%) (p=0.046)
Bear in mind that this, if expressed to 2 decimal places, is not significant. Huge SDs - beware drawing conclusions from this based on such a small sample size.
Line 204 'The first hypothesis is fully supported' This is an over-interpretation of the results
Line 205 'The sound control group can not be considered as significant' - what does this mean?
(2nd hypothesis: Orthopaedically healthy horses load and unload the extremities equally and regularly.)
Figure 1 Lame & non-lame limb of cases would be preferable to affected and non-affected
Line 209 'Lying behaviour also generally changes in lame horses, which is shown in Figure 3. They lie down more often, which is statistically significant (p = 0.025) 208 and supports the third hypothesis.
They lie down more often - did you measure the frequency of getting down & up or the total duration of time spent lying down or both? What does this refer to?
Line 210 Presumably this is text, not part of the legend for Fig. 2. The legend for Figure 2should provide more information
Figure 3
What is on the y axis?
It is extraordinary if non-lame horses were not lying down - that is not normal - there must be a reason
Please refer to other sleep studies
Discussion
Line 216 If the results are reported to 2 decimal places there is no significant difference between the lame & non-lame limbs - hypothesis 2 is not supported - although this may reflect the small sample size - but the sample is biased towards horses with upper levels of lameness grade
Line 220 Surely, if numbers had been bigger (what is the power of your study?) then the data should have been analysed to include lameness grade & duration of lameness & whether or not the horses were in work - all of which would potentially influence the results.
I teased out that less lame horses were less likely to rest the lame limb more than the non-lame limb - this should ideally be in the results
You probably also need to consider that a very lame horse may overload the non-lame limb predisposing to laminar pathology and thus contralateral limb discomfort which may alter weight bearing characteristics between the limbs
Lines 222 - 228 I think that this section on laterality is excessively long relative to its relevance to the current study. Relevant information could be expressed in a much more concise form
Line 234 onwards - this is relevant - but should be a separate paragraph & expressed more concisely
Line 245 But some lame horses lie down less because it is uncomfortable to get up & down - & in the long term suffer sleep deprivation.
I think that you need to look closer at the current literature relating to lying down and sleep
Line 246 'It was assumed that this was also reflected in the resting behaviour, but this could not be proven statistically. '
According to the methods you did not assess the effect of lameness grade - this is misleading
Line 249 I do not think that it would be ethically acceptable to induce lameness for a sufficient period of time to draw any conclusions that were relevant to a clinical scenario, particularly given that there is no shortage of horses with naturally occurring lameness
Line 254 'The degree of lameness reflects the amount of pain a horse is in' I don't think this is true - this depends on many factors - the type and duration of pain, the nature of pain, the underlying cause(s) and whether pain is indeed only unilateral. And of course recent work history, each horse's individual pain tolerance etc etc.
Line 256 s there any scientifically based valid information to support breed or type variability?
Line 260 ‘In the present study, attention was only paid to lame or not lame, which is why the 259 consideration of the different influence of the degree of lameness was not pursued further ‘- this seems to conflict with the inferences in the previous paragraph.
Line 264 But you have not presented any objective lameness data! – not even your inclusion criteria for lame or non-lame.
Line 266 There are a number of studies that indicate that even in foals asymmetry outside normal reference values can be measured which is not necessarily of clinical significance. Interpretation is not necessarily straightforward.
Objective gait analysis is not always better than subjective analysis; it measures asymmetry & may miss bilaterally symmetrically lame horses.
Line 271 However, this only applies to this study, if the lameness was so minor that 271 it was a question of whether the horse could be included or not
You have provided no evidence to support this statement
Line 273 How can you be sure that 11 hours is sufficient - you have not tested repeatability or longer time periods - and you have not idea what you might find with a group of horses in which a lower grade of lameness predominated. Moreover the AAEP scale is poor at discriminating between a mild consistently apparent lameness and a more severe consistently apparent lameness.
Line 276 If this information is to be useful you would have to study a much larger group of both lame & non-lame horses & assess repeatability of observations - ideally in home environments.
Line 280 ‘All measurements were taken in equine clinics and therefore in unfamiliar environments,’ - this should have been in the M&M
Line 286 You argued in the beginning that one of your objectives was to recognise the likely presence of lameness sooner rather than later - & if this is going to happen a horse has to be presented to a veterinarian for investigation. So the tool has to be utilisable in a home environment by the caregiver.
I am not at all clear what you mean by 'However, it would be interesting to be able to use the resting behaviour between, for example, diagnostic tests and local anaesthesia.'
Line 291 ‘If a procedure were to be developed that leads to faster availability of the evaluation, this could be a useful tool in the assessment of lameness.’
As already stated this is entirely reliant on the caregiver having an incentive to request veterinary investigation. So if you could show in a horse's home environment that there were changes in lying down behaviour between a horse being non-lame and then lame this might be valuable.
Or you need a much larger data set of non-lame horses and horses with a much wider range of grade of lameness.
Lines 293-299 This paragraph is not of direct relevance to the current study. Horses are prey animals and have evolved to conceal pain. Pain is a very subjective subject in even in people. To think we can analyse this in the horse is perhaps naive.
Line 304 What do you mean by 'at this time' ?- this tends to imply that a previous study may have involved the use of video recordings.
Line 308 Please refer to & study the paper of Torcivia & McDonnell
Line 317 I am not sure of the practical application of pressure mats in the scenario that you wish to investigate - which is patterns of shifting weight which does not necessarily reflect lameness & frequency and duration of lying down. Observation of the way in which horses get down & get up could also prove valuable - there are often changes in horses with chronic hindlimb lameness.
You have no indication of whether or not your results indicate that this information could facilitate early detection of lameness - the reader is provided with no information about the duration of lameness in the lame horses in the current study.
Line 331 'The early detection of potential lameness can be proven with this work,'
I don't think that you have provided any evidence to support this statement.
Conclusions
Line 335 'This work was carried out to investigate the early detection of lameness '
Your study design did not permit this
Comments on the Quality of English Language
There are quite a few instances (see my report) in which I cannot understand what the authors are trying to say. Much of the text could be expressed much more concisely.
Author Response
PLEASE SEE ATTACHEMENT WHERE THE ANSWERS ARE IN RED - OTHERWISE IT IS THE SAME
This could be a potentially interesting paper - but I think that the sample sizes are too small & the SDs too large to draw meaningful conclusions. There was no repeatability of the observations, to determine how consistently they occurred.
You are right that larger groups and different observation periods would have led to a clearer result. For practical reasons, the number of horses and the observation period were kept kind of low, as we wanted to show in a first step that our approach can work in principle. And we found statistically significant differences in the resting behavior of lame/painful horses! Follow-up studies with more horses and different observation periods would complete the picture we have drawn. For these reasons, we think that both the number and the observation period are sufficient for the first approach. In addition, the method should be used to monitor the horses so that deviations from the usual resting behavior can be detected.
There is inadequate description of the methods to be able to reproduce the study. There is overinterpretation of the results with potentially misleading conclusions.
Hopefully in the following the methods are more adequate and it will be more clear how to reproduce the study, because apart from analysing the data, it is an easy handling for the owner. And of course the owner has to start the measurements, when the horse is non-lame/normal/as always, so she/he can THEN spot differences for the individual own horse. Hopefully the results are more clear and the interpretation less misleading, so nobody can draw misleading conclusions anymore. Since our statistics indicate a significant difference, we do not believe this is an overinterpretation.
The authors stated that 'This work was carried out to investigate the early detection of lameness', however the design of the study does not permit this to be performed.
Please see it mentioned in the points below as this is mentioned multiple times during the work. In general, you are right with that and it has to be rewritten, which is now hopefully more clear.
The early recognition of lameness / musculoskeletal pain by a caregiver stimulating the request for veterinary investigation would certainly be of benefit. We know that riders and trainers are poor at recognising lameness based on numerous studies.
That would definitely be the aim of the study to be able to help all the mentioned people to recognize lameness. But not just in movement like described in previous literature, but also at rest. Therefore we tried to investigate in a study design in this direction. With monitioring the behaviour of resting hind limbs and laying down, we would like to show, that this can be a useful tool to monitor ones own horse at rest already and detect changes to the (for this particular horse) normal behaviour.
Some of the text reflects lack of adequate knowledge about the literature relating to equine lameness manifestations, behavioural adaptations to pain and sleep.
Please see all the comments / new texts and some new literature mentioned. I hope it draws a clearer picture of lameness, the use of accelerometers, behaviour, pain (within different breeds) and sleep. You will find the changes throughout the paper.
Many sentences could be constructed in a more succinct manner to both reduce the word count & to make the meaning clearer. The English requires substantial improvement.
Thank you for this comment, which is very useful for a non-Native speaker. The language was checked again by an experienced equine veterinary surgeon and Native speaker and I hope it meets your requirements in a better and more scientific way now.
In my opinion very substantial revision of the manuscript is required for a study which could only be regarded as a pilot study from which no definitive conclusions can be drawn.
You are right that larger groups and different observation periods would have led to a clearer result. For practical reasons, the number of horses and the observation period were kept kind of low, as we wanted to show in a first step that our approach can work in principle. And we found statistically significant differences in the resting behavior of lame/painful horses! Follow-up studies with more horses and different observation periods would complete the picture we have drawn. For these reasons, we think that both the number and the observation period are sufficient for the first approach. In addition, the method should be used to monitor the horses so that deviations from the usual resting behavior can be detected.
Title
‘and horses with unilateral hindlimb lameness’ would be a preferably wording
Please see the optimized title: Monitoring of sound horses and horses with unilateral hindlimb lameness at rest with the aid of accelerometers
‘at stance’ is misleading because you were evaluating their behaviour in a stable including lying down
Please see the optimized title: Monitoring of sound horses and horses with unilateral hindlimb lameness at rest with the aid of accelerometers
Abstract
Line 13 monitored not detected
Detected was changed for monitored
Line 14 please use correct anatomical nomenclature - not cannon bone
metatarsal region if a hindlimb
Cannon bone was replaced with correct anatomical nomenclature. Every time it was mentioned.
It should be made clear that both hindlimbs were instrumented (here & in the M&M )
Both hindlimbs were instrumented was added to the M&Ms and here.
Line 16 lying down
rest patterns
This was changed.
Line 18
‘Horses with unilateral hind lameness rest the affected leg significantly longer’
This will surely depend on the severity of the lameness
What was the statistical power of the study?
Here is the calculation of power with G-Power.
t tests - Means: Difference between two dependent means (matched pairs)
Analysis: A priori: Compute required sample size
Input: Tail(s) = One
Effect size dz = 0.62
α err prob = 0.05
Power (1-β err prob) = 0.95
Output: Noncentrality parameter δ = 3.3958799
Critical t = 1.6991270
Df = 29
Total sample size = 30
Actual power = 0.9525610
We estimated the group size using the G-Power software. The result was a total number of 30 horses (power 0.95). The power / group size of 30 horses is sufficient, as we could show statistical significant differences.
Line 20 If the result was expressed to 2 decimal places - which is appropriate - the result is not statistically significant (p=0.05).
To simplify the statistics, all horses were included and the resting relationship was determined (lame horses: healthy vs. lame or healthy horses: right vs. left) and the statistics were recalculated. First, we checked the data for normal distribution using a Kolmogorov-Smirnov test. Since the data were not normally distributed, we used a Mann-Whitney test to compare the groups. As a result, we were able to demonstrate statistically significant resting behavior (p=0.035 and p=0.04<0.05, respectively).
I suspect that your study sample is too small to draw meaningful results, & there is no repeatability over several days of observation. This cannot be considered anything other than a pilot study.
The power of 30 horses is sufficient, as we could show statistical significant differences.
‘11 hours was considered to be sufficient’ On what bases could this statement be made – you did not compare variable periods of assessment.
As you mentioned completely right, we don’t have a comparison between different periods of assessment. Due to that we changed the formulation into the following: The observation period was 11 hours.
Was there any human presence during this time?
Yes, there was a human (Anja Uellendahl) present for the first few minuets, but this doesn’t affect the study. The sensors are absolutely capable to detect the different postures by their own. Which is then repeatable.
Was video surveillance also used? - & if not why not?
In this study video surveillance was not used, because the sensors are able to show the different postures with the created values and the changes amongst them. In a pre study, which was Anja Uellendahls diploma thesis (= Uellendahl 2019) (in German) to end the studies, we tested, if the sensors are able to differ between the different resting patterns. To control that, we had video surveillance to see, when the horse rests its legs or lays down. If the behaviour occurs, we had a look at the data and the changes in the values from the sensors and after that pre study we knew, that it is possible to detect resting behaviour just with the sensors and without video surveillance.
What does ' the total of the lame horses' mean?
The total of lame horse means all of them, but it is written like this now: the group of lame horses.
Line 21 lay down for significantly longer
This line was changed. Thank you.
This will depend on degree of discomfort - some lame horses have reduced lying down & experience sleep deprivation.
Please see the new literature mentioned all over here and the introduction about sleep in horses.
Introduction
Line 28 horses' behaviour
This was changed.
Accelerometers only tell you a limited amount about behaviour - when & to want extent a horse is moving.
The word behaviour has been replaced in the whole document to posture / rest / resting pattern. Behaviour is a huge and more complicated topic, which can’t be fully assessed by accelerometers as you said right. The Accelerometers create values and their changes allow us to detect a different posture. It is a very practicable and simple solution to use the accelerometers for a reference to the deviation from the vertical acceleration due to gravity.
Line 28 its creation - what does its refer to?
Unfortunately this can’t be seen in line 28.
Line 58 what does 'in this study ' refer to?
This refers to the study which was connected with the number in the brackets.
Line 61 You need to look at Torcivia & McDonnell's work
Thank you for this recommendation, please find it added in the paper.
Line 81 lay down
That is changed. Thank you.
Materials and Methods
Line 88 Were these horses in their home environments or at a clinic? If at a clinic for how long had they been there before acquisition of the results? This information should be in the M&M.
These horses have been tested in a clinic. The lame horses mostly have been in the clinic just one day before and were tested in the first night after arriving to the clinic, because we couldn’t wait longer (for the horses to maybe settle in more into the unknown environment) with further lameness investigations. The non-lame horses have also been tested in their first or second night, so very early during their stay. This information was added in the M&M.
How may these factors have influenced the results?
Horses in a clinical environment may be more nervous. The adrenalin maybe helps them to deal with pain and show less resting behaviour. This might influence the results more than it probably would in a home environment. This was also added in the discussion.
Line 89 what is meany by 'the revised criteria'?
This basically means lame or non-lame, but it was removed to reduce confusion.
Line 92 Determined how? Under what circumstances were the non-lame horses examined. Were these horses in regular work? how can you say 'no orthopaedic disease'? They could have had clinically silent radiological abnormalities consistent with osteoarthritis.
The non-lame horses were examined in the same way like the lame ones. All of them have been in some kind of work. The non-lame have been ridden, but not all of them on a sports level, but this wasn’t a criteria anyway. The ‚no orthopaedic disease‘ was replaced constantly with non-lame horses or similar. You are right, we can’t tell about clinically silent abnormalities, which is the reason for the changes to non-lame.
Non-lame on day of examination and in regular work, working willingly?
Yes, this was changed and added in a more comprehensible way.
Line 94 Under what circumstances were the horses evaluated to determine they were grade 1/5 on the AAEP scale?
All of them have been walked and trotted on a hard ground in straight line. They have been lunged on a soft surface in both directions as well as in all 3 gaits. Horses which were barely visibly lame and which were detected by the experienced veterinarian and the Equinosis lameness Locator, have been graded low. This has been added to the methods section.
Line 97 How can you say with certainty that a horse is unilaterally lame unless the horse is blocked to abolish the lameness and the horse does not switch lameness to the contralateral limb?
Due to the fact, that the lame horses have been real clients, they underwent a lameness evaluation including blocks. In non of the blocked horses the lameness switched to an other leg. A few horses, which were brought to the hospital with a lameness underwent the physical evaluation (trotting and lungeing) on day one, but have been blocked the next day. Anyway, the data has been collected in their first night at the hospital and used the data after they had positiv blocks on the next day.
Table 1 All tables & figures should have comprehensive legends so that these can be read independently from the text
May we asked you politely what is not comprehensive in Table 1? How would you improve the already detailed headlines in a better way? We gave it a try like this: Detailed list of the used horses with their breed, year of birth, sex, group (non-lame, lameness with side and degree) and the test date.
How were the horses selected?
The lame horses have been selected because of the case load of the hospital. As soon as there was a lameness appointment booked in, the client was asked to participate in the study. We didn’t proactively look for lame horses, but used the existing case load. Non-lame horses, which have been in the hospital for a different disease (like endoscopies, gastroscopies, eye treatments or e.g. mass removals before surgery or simply companion horses for other sick horses) have been selected because of the absence of lameness. They underwent the trotting and lungeing like the lame horses.
There are a high proportion of horses with grade 4/5 lameness - that is severe and atypical of the sports horse.
That is a good point. There was a wide range of different types of horses on different levels of activity. Almost all of them have been leisure horses and not sport horses. That is the reason why we also had severe lame horses.
Please check all English spellings of the breeds and define any abbreviations e.g., PRE
This is checked and all breeds have been changed in an English spelling.
Line 108 metacarpal or metatarsal region? Not cannon bone
This was changed over the whole paper.
Which limb(s)??
All limbs.
Line 116 why not lying down?
Because it took too long and we wanted to avoid irritation about a human being present during the night. We thought horses would maybe not lie down then. We wanted to create the most possible way of a „normal routine“ and didn’t want to disturb the horses by standing next to the box. Also the values for lying down are absolutely different to a standing position and could be filtered out easily later.
Where was the person - what was their experience - what did they record - over what time period were horses observed?
The person was standing outside the box. The horses have been observed until they showed the defined behaviour once. The time was written down in minutes how long the behaviour was showed an then we knew exactly by time which values in the sensors excel file were connected to the behaviour of this particular horse. This was done for personal reasons, just in case every horse rests its leg differently or more subtle. We (because we didn’t want to miss out on data, we potentially could have uses later) didn’t need to double check, because the sensors are absolutely capeable of showing the resting patterns. We just wanted to see, if the individual horse is tolerating the sensors.
Line 118
‘Each time, another horse was in the same stable’
What does this mean?
This means, that the horse was never alone in the stable. It had its own box, but there was always a second horse around it. Either as a direct neighbour or across the aisle in a different box. This was changed to: Every horse had a companion in the box next to it or across the aisle.
Line 121 So 2 per horse, one on each hindlimb ?
No. Please see the question below.
Or (line 134) So were 4 accelerometers used per horse, one on each limb?
Yes. 4 have been used, but 2 have been looked at. Just the two on the hindlimbs. We prefered to collect the data of all legs, but as soon as we have been sure to just look at the hind limbs, we ignored the data of the two fronts.
Line 143 Avoid repetition!
Okay. Hopefully the repetition you meant exactly in this line was deleted.
Line 150 ‘this had the greatest deflection and the highest sensitivity’ - based on this work or previous work??
Not clear!
It was the best way to detect the angle to the vertical direction (nearly 90 degree for lay down on all 4 sensors and a certain difference between the left and the right hind limb during resting one limb). We use SPSS 29 to calculate the statistical comparisons. At first, we checked the data for normal distribution with a Kolmogorov-Smirnov Test. The data were not normal distributed therefore, we used a Mann-Whitney Test to compare the groups. Additional we calculated the Spearman-Rho correlation coefficient between the data. This was added to the text.
Line 155 within not among
This was changed.
Line 167 'certain ratios of the values were exceeded ' - what does this mean?
So which horses were excluded
How many lame & non-lame horses?
Any why do the tables include these excluded horses?
To simplify the statistics, all horses were included, and the resting relationship was determined (lame horses: healthy vs. lame or healthy horses: right vs. left) and the statistics were recalculated. First, we checked the data for normal distribution using a Kolmogorov-Smirnov test. Since the data were not normally distributed, we used a Mann-Whitney test to compare the groups. Additional we calculated the Spearman-Rho correlation coefficient between the data. Please see herefore the new table.
How were the non-lame horses determined to be non-lame? Under what circumstances were they evaluated?
They underwent the same lameness evaluation like the lame horses, which was mentioned above. They were equipped with the same lameness Locator system and also with the eye of an experienced veterinarian. And Anja Uellendahl as a new graduate with high motivation in lameness. See also new Table (2&3) .
What was the duration of lameness in the lame horses?
This varied in each case. Sublte lameness have been going on for longer then more severe lamenesses. This could be explained by that the owners waited longer or didn’t see the lameness early. Unfortunately we can’t tell the exact time for each horse anymore. Lameness score see Tabe 1.
Line 172 Why was log transformation performed?
To simplify the statistics, all horses were included, and the resting relationship was determined (lame horses: healthy vs. lame or healthy horses: right vs. left) and the statistics were recalculated. No transformation was performed.
Results
Table 2 All tables & figures should have comprehensive legends so that these can be read independently from the text
Please see the changed legends.
n=9
N=10 in original definition
3 & 26 outliers?
Table 3 Why are horses 6 & 15 outliers?
Large SD
n=8
6 grade 3/5
15 grade 1-2/5
To simplify the statistics, all horses were included, and the resting relationship was determined (lame horses: healthy vs. lame or healthy horses: right vs. left) and the statistics were recalculated. We now present the new statistics, which we hope are clear and easy to understand.
How can a horse be between grades?
By refining the score, it was more practicable for the vets because a horse can be between grades if both definitions fit partially. If it is too lame for the lower grade and not consistently lame for the next grade. An experienced vet and Anja Uellendahl graded both and sometimes they had to do the mean.
Table 4
16 & 21 outliers
large SD
Why are there 13 horses when you said in the text that there were 12 horses with RH lameness?
In table 1 horse 7 is listed as non-lame
To simplify the statistics, all horses were included, and the resting relationship was determined (lame horses: healthy vs. lame or healthy horses: right vs. left) and the statistics were recalculated.
16 grade 3/5
21 grade 2/5
Line 189 Why different for 3 & 26?
To simplify the statistics, all horses were included, and the resting relationship was determined (lame horses: healthy vs. lame or healthy horses: right vs. left) and the statistics were recalculated.
Line 196 'The total of the lame horses (13±11%) 196 lay significantly longer than sound horses (3±6%). '
What does this mean - that only 3 non-lame horses lay down at all?
Yes, in our study we found that healthy horses rarely lie down. This behavior is clearly associated with lameness..
Line 200 sound group, healthy group - please be consistent in terminology
Non-lame may be better
We tried to alter the definitions for a better reading / understanding, but we changed now every terminology to non-lame.
The Results need to be expressed more succinctly
Okay. Please see the revised results, which are hopefully expressed more succinctly.
Please express percentage results to no more than 1 decimal place
This was changed to 1 decimal place in the whole document.
Line 202 What does 'settle at a similar time in %' actually mean?
It was ment, that it is 50/50 between left and right leg in the non-lame control group, when you look at the mean values. But there is a high individual difference between the horses. This was now changed to: The mean values show an almost 50% / 50% distribution between the legs. This is changed in the text like this: It can be seen that, as in the non-lame group the individual variance can be high, but in the mean the percentage distribution approximates 50% / 50%.
This describes, as an example, how the text could be shorter: The lame hind limb was rested for longer (61.8% ±25.3%) than the non-lame limb (38.2%±25.3%) (p=0.046)
Thank you very much for giving this example. We changed it to lame and non-lame / sound leg of the lame group of horses.
Bear in mind that this, if expressed to 2 decimal places, is not significant. Huge SDs - beware drawing conclusions from this based on such a small sample size.
To simplify the statistics, all horses were included and the resting relationship was determined (lame horses: healthy vs. lame or healthy horses: right vs. left) and the statistics were recalculated. First, we checked the data for normal distribution using a Kolmogorov-Smirnov test. Since the data were not normally distributed, we used a Mann-Whitney test to compare the groups. As a result, we were able to demonstrate statistically significant resting behavior (p=0.035 and p=0.04<0.05, respectively).
Line 204 'The first hypothesis is fully supported' This is an over-interpretation of the results
Since our statistics indicates a significant difference, we do not believe this is an over-interpretation.
Line 205 'The sound control group can not be considered as significant' - what does this mean?
(2nd hypothesis: Orthopaedically healthy horses load and unload the extremities equally and regularly.)
This was a mistake. There is no significant difference between the hind legs in the sound = new non-lame group. Which is exactly what we wanted to show. It is written differently now. See new: The non-lame control group can also be considered as significant (p = 0.751), which does support the second hypothesis. Additionally, the resting relation was calculated and compared between lame and non-lame horses (p=0.035).
Figure 1 Lame & non-lame limb of cases would be preferable to affected and non-affected
In this case we decided for affected and non-affected to minimize confusion. Talking about lame and non-lame horses, we wanted to have a different term, when we were talking about the legs within the lame group of horses. But it has been changed to lame and non-lame leg of the lame group of horses.
Line 209 'Lying behaviour also generally changes in lame horses, which is shown in Figure 3. They lie down more often, which is statistically significant (p = 0.025) 208 and supports the third hypothesis.
They lie down more often - did you measure the frequency of getting down & up or the total duration of time spent lying down or both? What does this refer to?
In this case we indeed measured both. Just the total duration of time spent lying down was used for the statistics, not the frequency as well.
Line 210 Presumably this is text, not part of the legend for Fig. 2. The legend for Figure 2should provide more information
We added more text to the legend like this: Comparison of the percentage distribution of the hindlimb resting time between the lame and non-lame group. And within these groups between the affected and not affected leg within the lame group, respectively left and right within the non-lame group. ---- And yes that originally was text and we don’t know how it moved into the legend, because it was fine, when it was uploaded.
Figure 3
What is on the y axis?
This is the Relative Resting Time in %, but unfortunately misses in the figure, so this was added, thank you.
It is extraordinary if non-lame horses were not lying down - that is not normal - there must be a reason
Unfortunately we can’t tell you the reason. It might be the hospital environment. It was always quiet over night, no emergencies or people who have been around, checking the horses. But maybe horses didn’t like to lay down there.
Please refer to other sleep studies
Following studies were added:
- Greening, L., & McBride, S. (2013). A Review of Equine Sleep: Implications for Equine Welfare.
- Kelemen, Z., Grimm, H., Long, M., Auer, U., & Jenner, F. (2020). Recumbency as an Equine Welfare Indicator in Geriatric Horses and Horses with Chronic Orthopaedic Disease.
- Oliveira, T., Santos, A., Silva, J., Trindade, P., Yamada, A., Jaramillo, F., Silva, L., & Baccarin, R. (2021). Hospitalisation and Disease Severity Alter the Resting Pattern of Horses.
- Dallaire, A. (1986). Rest behavior.
- Clothier, J., Small, A., Hinch, G., Barwick, J., & Brown, W. Y. (2010). Using Movement Sensors to Assess Lying Time in Horses With and Without Angular Limb Deformities.
- Dallaire, A., & Ruckebusch, Y. (1974). Sleep patterns in the pony with observations on partial perceptual deprivation.
Discussion
Line 216 If the results are reported to 2 decimal places there is no significant difference between the lame & non-lame limbs - hypothesis 2 is not supported - although this may reflect the small sample size - but the sample is biased towards horses with upper levels of lameness grade
This value is very close to the limit, so we think it was justified to point it out. All numbers have been changed to maximum only one decimal place.
I teased out that less lame horses were less likely to rest the lame limb more than the non-lame limb - this should ideally be in the results
Yes, here is the statistical proof. See tables and figures.
|
Teststatistikena |
||||
|
|
Vetscore |
Lamenesslocator |
Restrelation |
laytime |
|
Mann-Whitney-U-Test |
,000 |
49,000 |
52,000 |
44,000 |
|
Wilcoxon-W |
55,000 |
104,000 |
262,000 |
99,000 |
|
Z |
-4,532 |
-2,245 |
-2,114 |
-2,527 |
|
Asymp. Sig. (2-seitig) |
<,001 |
,025 |
,035 |
,011 |
|
Exakte Sig. [2*(1-seitige Sig.)] |
<,001b |
,024b |
,035b |
,013b |
|
a. Gruppenvariable: Gruppe |
||||
|
b. Nicht für Bindungen korrigiert. |
||||
Line 220 Surely, if numbers had been bigger (what is the power of your study?) then the data should have been analysed to include lameness grade & duration of lameness & whether or not the horses were in work - all of which would potentially influence the results.
The power of 30 horses is sufficient, as we could show statistical significant differences. But. Of course it would be great to include the degree of lameness, the duration and the work of horses, but in this study we tried to just analyse, if the lame horses rested the affected leg longer. I will add these three points in the discussion, like this: Due to the smaller number of horses, no statement was made about the connection to the degree of lameness, the duration of lameness and the intensity of the horse's normal training. This was not taken into account in the present study, but would be a good investment for future studies.
|
Teststatistikena |
||||
|
|
Vetscore |
Lamenesslocator |
Restrelation |
laytime |
|
Mann-Whitney-U-Test |
,000 |
49,000 |
52,000 |
44,000 |
|
Wilcoxon-W |
55,000 |
104,000 |
262,000 |
99,000 |
|
Z |
-4,532 |
-2,245 |
-2,114 |
-2,527 |
|
Asymp. Sig. (2-seitig) |
<,001 |
,025 |
,035 |
,011 |
|
Exakte Sig. [2*(1-seitige Sig.)] |
<,001b |
,024b |
,035b |
,013b |
|
a. Gruppenvariable: Gruppe |
||||
|
b. Nicht für Bindungen korrigiert. |
||||
You probably also need to consider that a very lame horse may overload the non-lame limb predisposing to laminar pathology and thus contralateral limb discomfort which may alter weight bearing characteristics between the limbs
This was added in the discussion: Horses with a higher lameness degree may overload the contralateral hind limb and discomfort may occur. How this influences the actual resting behaviour would be a good investment for future studies.
Lines 222 - 228 I think that this section on laterality is excessively long relative to its relevance to the current study. Relevant information could be expressed in a much more concise form
The section about laterality was shortened. Also the whole paper underwent an English review by an experienced equine veterinarian and Native speaker.
Line 234 onwards - this is relevant - but should be a separate paragraph & expressed more concisely
A seperate paragraph was added for this.
Line 245 But some lame horses lie down less because it is uncomfortable to get up & down - & in the long term suffer sleep deprivation.
I think that you need to look closer at the current literature relating to lying down and sleep
As mentioned and listed above, there is now more literature. You can find more literature and citations in the text and references.
Line 246 'It was assumed that this was also reflected in the resting behaviour, but this could not be proven statistically. '
According to the methods you did not assess the effect of lameness grade - this is misleading
The lameness locator Data was used to now create statistics to the lameness degree to assess the effect of the lameness degree on the resting pattern (resting leg and laying down). It was added to the results and M&Ms. Please see table 2 and 3.
Line 249 I do not think that it would be ethically acceptable to induce lameness for a sufficient period of time to draw any conclusions that were relevant to a clinical scenario, particularly given that there is no shortage of horses with naturally occurring lameness
This was just another idea. The future of gait analysis lies in non-invasive technologies like wearable sensors, high-speed cameras, and machine learning algorithms. These tools can monitor naturally occurring lameness, providing valuable data without ethical concerns. Big data and AI can further enhance understanding by analyzing large datasets, leading to more accurate diagnoses and treatments. In summary, while induced lameness models have advanced gait analysis, they pose ethical challenges. The future will focus on non-invasive methods that uphold animal welfare while continuing to innovate in equine gait analysis.
87 Results from 2019 to 2024 (from 02.08.2024)
Line 254 'The degree of lameness reflects the amount of pain a horse is in' I don't think this is true - this depends on many factors - the type and duration of pain, the nature of pain, the underlying cause(s) and whether pain is indeed only unilateral. And of course recent work history, each horse's individual pain tolerance etc etc.
This was changed to: This is influenced by many factors like the duration of lameness and its cause and so the duration, cause, nature and the individual tolerance of pain in each horse. Also if the horse has a change in its behavior during training.
Line 256 s there any scientifically based valid information to support breed or type variability?
In the mentioned website was written, that there are differences. Recognition and Management in Horses | News | Merck Equine. Available online: https://www.merck-animal-health-476 equine.com/news/article/34
Also please see a bit more of research and more literature added, which you can see here (and please also see the new passages in the text itself):
Gleerup, K. B., Forkman, B., Lindegaard, C., & Andersen, P. H. (2015). An equine pain face. Veterinary Anaesthesia and Analgesia, 42(1), 103-114.
Dalla Costa, E., Minero, M., Lebelt, D., Stucke, D., Canali, E., & Leach, M. C. (2014). Development of the Horse Grimace Scale (HGS) as a pain assessment tool in horses undergoing routine castration. PLoS One, 9(3), e92281.
Lesimple, C., Fureix, C., De Margerie, E., Sénèque, E., Menguy, H., & Hausberger, M. (2016). Towards a pain scale in horses based on facial expressions (EquiFACS): A pilot study. PLoS One, 11(8), e0159716.
Dyson, S., Berger, J., Ellis, A. D., & Mullard, J. (2017). Development of an ethogram for a pain scoring system in ridden horses and its application to determine the presence of musculoskeletal pain. Journal of Veterinary Behavior, 19, 89-101.
Line 260 ‘In the present study, attention was only paid to lame or not lame, which is why the 259 consideration of the different influence of the degree of lameness was not pursued further ‘- this seems to conflict with the inferences in the previous paragraph.
Yes, that is true, thank you. We added some more statistics out of the lameness data of the lameness locator. So please see the new part in the discussion about this.
Line 264 But you have not presented any objective lameness data! – not even your inclusion criteria for lame or non-lame.
The objective lameness data is now presented by the new and added statistics with new text out of the data of the lameness locator. The inclusion criteria for non-lame horses is the absence of a lameness. The inclusion criteria for lame horses is the presence of a unilateral hindlimb lameness.
Line 266 There are a number of studies that indicate that even in foals asymmetry outside normal reference values can be measured which is not necessarily of clinical significance. Interpretation is not necessarily straightforward.
Objective gait analysis is not always better than subjective analysis; it measures asymmetry & may miss bilaterally symmetrically lame horses.
Hopefully this is more straightforward now. There always had been an experienced equine veterinarian next to Anja Uellendahl. Anja Uellendahl also looked at the horses (with less experience, but high motivation for lameness evaluations and being as precise as possible). This was combined with the Lameness Locator. We think, if one wants to use these technical aids is like so much an individual decision.
Line 271 However, this only applies to this study, if the lameness was so minor that 271 it was a question of whether the horse could be included or not
You have provided no evidence to support this statement
This sentence was deleted to reduce the confusion.
Line 273 How can you be sure that 11 hours is sufficient - you have not tested repeatability or longer time periods - and you have not idea what you might find with a group of horses in which a lower grade of lameness predominated. Moreover the AAEP scale is poor at discriminating between a mild consistently apparent lameness and a more severe consistently apparent lameness.
This was changed in the way, that the 11 hours were mentioned, but not interpreted anymore. This can also be something for future studies. Like this: All horses have been measured in consecutive 11 hours. If shorter time periods, which might be more realiable for horse owners, are functional, is worth to be looked at in future studies. We decided to use the AAEP scale, because this is what we got taught at university and is the most familiar scale to us all. There shouldn’t be any investigation about how good or bad the scale is as long as we consistently used the same scale.
Line 276 If this information is to be useful you would have to study a much larger group of both lame & non-lame horses & assess repeatability of observations - ideally in home environments.
Of course it would be amazing to have all these factors included. You are right that larger groups and different observation periods would have led to a clearer result. For practical reasons, the number of horses and the observation period were kept kind of low, as we wanted to show in a first step that our approach can work in principle. And we found statistically significant differences in the resting behavior of lame/painful horses! Follow-up studies with more horses and a different observation periods would complete the picture we have drawn. For these reasons, we think that both the number and the observation period are sufficient for the first approach. In addition, the method should be used to monitor the horses so that deviations from the usual resting behavior can be detected. And it is absolutely repeatable, even if you can absolutely improve (via App maybe?) how the data can be processed quicker.
Line 280 ‘All measurements were taken in equine clinics and therefore in unfamiliar environments,’ - this should have been in the M&M
This was added to the M&M.
Line 286 You argued in the beginning that one of your objectives was to recognise the likely presence of lameness sooner rather than later - & if this is going to happen a horse has to be presented to a veterinarian for investigation. So the tool has to be utilisable in a home environment by the caregiver.
As long as the owner considers its horse as normal and therefore not lame, the tool is utilisable at home. It is easy to use, but the analysis of the data has to be more easy and quicker. This is just doable by someone who can read the values. But for sure there are many systems which can work with apps on the phone and be able to analyse the data during the measurement process. The owner knows through repeated measurements of her/his own individual horse what the percentage distribution between the limbs of the individual horse is and can therefore realise any changes in the resting behaviour sooner rather then later.
I am not at all clear what you mean by 'However, it would be interesting to be able to use the resting behaviour between, for example, diagnostic tests and local anaesthesia.'
That means, that it would be interesting as soon as there is the possibility of not needing 11 hours to measure (which we not know yet as you said right), if the resting behaviour of the horse even changes between diagnostic anaesthesias (= blocks). E.g. the horse rests ist limbs 70%-30% but after the first block it is 60%-40% which might show an improvement. But until then there are many steps which have to be investigated beforehand.
Line 291 ‘If a procedure were to be developed that leads to faster availability of the evaluation, this could be a useful tool in the assessment of lameness.’
As already stated this is entirely reliant on the caregiver having an incentive to request veterinary investigation. So if you could show in a horse's home environment that there were changes in lying down behaviour between a horse being non-lame and then lame this might be valuable.
Or you need a much larger data set of non-lame horses and horses with a much wider range of grade of lameness.
In this case I meant the processing of the data, which takes hours for one human induvidual and that we don’t know, if e.g. 1 hour of observation would be enough (instead of 11). Not the practibility for an owner. As long as the owner considers its horse as normal and therefore not lame, the tool is utilisable at home. It is easy to use, but the analysis of the data has to be more easy and quicker. This is just doable by someone who can read the values. But for sure there are many systems which can work with apps on the phone and be able to analyse the data during the measurement process. The owner knows through repeated measurements of her/his own individual horse what the percentage distribution between the limbs of the individual horse is and can therefore realise any changes in the resting behaviour sooner rather then later. Here will be more possibilities for research in this field.
Lines 293-299 This paragraph is not of direct relevance to the current study. Horses are prey animals and have evolved to conceal pain. Pain is a very subjective subject in even in people. To think we can analyse this in the horse is perhaps naive.
You asked about having a closer look at the pain management, e.g. in different breeds. That is what we now did and changed this paragraph. Hopefully this has more relevance now. If you want us to talk about pain in a complete different way to what it was changed now, we would appreciate help. Thank you.
Line 304 What do you mean by 'at this time' ?- this tends to imply that a previous study may have involved the use of video recordings.
Yes. To finish the veterinary studies we have to write a Diploma thesis. In this we used video surveillance to be able to see, if the used sensors are able to detect the described resting behaviours. Unfortunately this is not published, so the sentence will be rewritten. Apart from that, we added some sentences about accelerometer in general and hope, that it is less urgent or there is less need of having the diploma thesis to understand the current study.
Line 308 Please refer to & study the paper of Torcivia & McDonnell
Done.
Line 317 I am not sure of the practical application of pressure mats in the scenario that you wish to investigate - which is patterns of shifting weight which does not necessarily reflect lameness & frequency and duration of lying down. Observation of the way in which horses get down & get up could also prove valuable - there are often changes in horses with chronic hindlimb lameness.
This was also just an idea, but the part with the pressure mats will be deleted and the observation point about how horses get up and down was added. But this would be just visuable with video surveillance, which was not existent in this study.
You have no indication of whether or not your results indicate that this information could facilitate early detection of lameness - the reader is provided with no information about the duration of lameness in the lame horses in the current study.
This has to be formulated differently throughout the whole paper to not mislead. More as an prospective hope or wish. And as already mentioned, it is importent, that the owner knows the individual resting pattern of the non-lame/normal horse and measures this until they spot a difference in the pattern of their own horse.
Line 331 'The early detection of potential lameness can be proven with this work,'
I don't think that you have provided any evidence to support this statement.
Please see the point above.
Conclusions
Line 335 'This work was carried out to investigate the early detection of lameness '
Your study design did not permit this
Please see the points above.
Comments on the Quality of English Language
There are quite a few instances (see my report) in which I cannot understand what the authors are trying to say. Much of the text could be expressed much more concisely.
The whole paper underwent a revision of an experienced equine veterinarian who is also a Native speaker.

Reviewer 2 Report
Comments and Suggestions for Authors
General comments
Dear Authors,
Lameness is a term used to describe an uneven gait caused by unequal weight bearing on one or more limbs due to pain. Gait unevenness, which means that it is generally assessed in movement unless the horse does not put any weight on the limb at all. However, early diagnosis is not needed in this case (see: American Association of Equine Practitioners. LAMENESS EXAMS: Evaluating the Lame Horse).
The Introduction section requires a complete rewrite. It should be a substantive introduction to the topic covered by the research, but at the moment it is quite a loose collection of non-specific statements, including general motivations, too-large generalizations, and data indicated to be provided in the M&M section. See detailed comments, with suggestions on how you can improve your work.
The main assumption regarding the detection of lameness in a standing position is incorrect, and the (more correct) measurement of loading/unloading time presented later requires explanation and clarification.
The aim and hypotheses of the work require improvement (see detailed comments).
Your work focuses on determining the uneven load on the limb at rest, not in movement. Which is not the same as early detection of lameness. You cannot confuse these terms. I believe that the research you have done is valuable and suitable for publication, but not in this form.
Unfortunately, from a veterinary point of view, the work requires a huge reconstruction, rethinking, and supplementing a huge amount of medical data influencing the measurement results. Please consult with an experienced veterinarian mentioned in L 94 and together rebuild the assumptions of the work so that they are clinically correct.
At this point, unfortunately, I have to recommend rejection of this article, but I will be happy to see it again after resubmission.
Detailed comments
L 5-8 Add references
L 9-24 Correct the abstract section by removing names of subsections (Objective, Study design and animals: Comparative study, Methods, Results, Conclusions and clinical relevance). Please, follow the guidelines available at https://www.mdpi.com/authors/layout)
L 28-31 Please, rewrite to show the background of your study, not general motivations. The introduction section aims to thoroughly introduce the reader to the topic covered by the research and how it is done is important.
L 29, L 43, L 89, L 138 Correct the way you cite references in the main text.
L 32-33 "The literature increasingly describes how various technical aids are used in lameness diagnostics [2]." I agree. However, it is a general statement. It would be more beneficial to tell the reader what various technical aids have been recently used in lameness diagnostics. See that the reference 2 is a short Letter/Commentary, not a systematic review.
L 33 Remove "According to previous studies". Everything you support with references is according to previous studies.
L 34-35 This is information that should be in the M&M section.
L 35 Remove "This has already been mentioned in previous publications". Everything you support with references is previously published.
L 37-38 "The recognition of lameness is an important point no matter in what way one deals with horses." What do you want to say with this sentence? Please remember that you are writing a scientific publication.
L 39 "naming lameness"? Do you mean clinically determining the degree and type of lameness, and limb affected by pain (of which lameness is a symptom) or identifying the cause of pain (diagnosis of lameness)? Support with appropriate reference.
L 42-44 See general comments and explain to me how you want to detect lameness in a standing position.
L 47-50 This should not be part of the introduction section.
L 51 " detect lameness already at rest" I am sorry, but you cannot detect lameness (uneven gait) at rest.
L 52 You may assess limb loading/unloading and limb loading/unloading time, but this is not a detection of lameness.
L 53-54 Be precise in what and how you quote, as references 10-12 do not describe the detection of lameness at rest.
L 60 Remove generalization " which would be contrary to the hypothesis put forward."
L 68 You are examining the loading/unloading of a limb, which may or may not be related to pain in that limb, not lameness.
L 70 What do you mean by "mild lameness"? What degree is it on the 0-5 AAEP scale? Which of these degrees do you think is objectively difficult to detect?
L 71 Since you claim to study horse behavior, what ethograms were assessed and how?
L 74 How do you know that the posture behavior you are examining is due to pain? What pain? How severe? In what area?
L 76 The pain you assume is hypothetical. Have you performed diagnostic anesthesia on the affected limb to see if the symptoms/ posture behavior will stop or change after the pain is relieved?
L 81 Have you ruled out any other factors influencing whether the horse lies down or not?
L 89 What criteria? Describe clearly and legibly the criteria for including and excluding horses from the study.
L 93 The basis for determining lameness is a clinical examination, not the results of Lameness Locator measurements. Please describe the diagnostic procedure. Further on, you refer to the AAEP system, but you do not say anything about the study protocol.
l 98-99 What do you mean by" In addition, the lameness must be recognizable to the Equinosis® Lameness Locator"?
L 100 Were the mares in heat or foal during the examination?
L 102-106 Was this the horses' natural habitat? Were they moved to a homogeneous environment? Were the studies conducted in homogeneous conditions?
The further part of the work can be assessed only after the assumptions, goal, hypothesis and the first part of the methodology related to the animal experimental system have been improved.
As for prospective experimental studies, a lot of information that is important from the point of view of study design is missing, and the way of presenting what is collected is poor.
Author Response
PLEASE SEE ATTACHEMENT WITH ANSWERS IN RED; REST IS THE SAME.
Dear Authors,
Lameness is a term used to describe an uneven gait caused by unequal weight bearing on one or more limbs due to pain. Gait unevenness, which means that it is generally assessed in movement unless the horse does not put any weight on the limb at all. However, early diagnosis is not needed in this case (see: American Association of Equine Practitioners. LAMENESS EXAMS: Evaluating the Lame Horse).
The early detection is a hope for the future by knowing the own horses resting patterns in the way of the distribution in resting its hind limbs and by the time which is spent lying down. As soon as each individual owner knows that, they can see changes in the distribution early before the horse maybe even shows lameness. The lameness has to be present (or not present) to be suitable and divided in two groups in this study. The lameness assessment was in movement. Walking and trotting on hard ground in a straight line and on a lunge on soft ground in both directions in all three gaits. This has been added to the text.
The Introduction section requires a complete rewrite. It should be a substantive introduction to the topic covered by the research, but at the moment it is quite a loose collection of non-specific statements, including general motivations, too-large generalizations, and data indicated to be provided in the M&M section. See detailed comments, with suggestions on how you can improve your work.
We will use the detailed comments to comment on this statement. Thank you.
The main assumption regarding the detection of lameness in a standing position is incorrect, and the (more correct) measurement of loading/unloading time presented later requires explanation and clarification.
Okay. Thank you for your comment. Hopefully from now on it is more clear throughout the whole paper.
The aim and hypotheses of the work require improvement (see detailed comments).
We will use the detailed comments to comment on this statement.
Your work focuses on determining the uneven load on the limb at rest, not in movement. Which is not the same as early detection of lameness. You cannot confuse these terms. I believe that the research you have done is valuable and suitable for publication, but not in this form.
That is true. We never wanted to assess the horses in movement, but at rest. We are sorry, if that is not clear. The early detection is a prospective hope by knowing the individual resting pattern of the horse and then deviations of that.
Unfortunately, from a veterinary point of view, the work requires a huge reconstruction, rethinking, and supplementing a huge amount of medical data influencing the measurement results. Please consult with an experienced veterinarian mentioned in L 94 and together rebuild the assumptions of the work so that they are clinically correct.
We would like to comment on this, but we think you have to read the Article again to see all the changes and if you like them better now. We can’t give you detailed changes here in our comment because of the severity of this review, but please have a look at the rewritten passages. Thank you.
At this point, unfortunately, I have to recommend rejection of this article, but I will be happy to see it again after resubmission.
Okay thanks.
Detailed comments
L 5-8 Add references
If you mean by references, mentioning the University etc., then this was removed by sensors themselves, because it was definately included in my firstly uploaded version. And Sensors does have this Information and can add it, if they want it.
L 9-24 Correct the abstract section by removing names of subsections (Objective, Study design and animals: Comparative study, Methods, Results, Conclusions and clinical relevance). Please, follow the guidelines available at https://www.mdpi.com/authors/layout)
Thanks for providing the guidelines. The headlines have been removed and the abstract is now one text block.
L 28-31 Please, rewrite to show the background of your study, not general motivations. The introduction section aims to thoroughly introduce the reader to the topic covered by the research and how it is done is important.
Please see the rewritten paragraphs.
L 29, L 43, L 89, L 138 Correct the way you cite references in the main text.
This citation style was used especially for Sensors by a programm called Citavi and after downloading the citation style. Could you please tell us exactly, what is wrong with it?
L 32-33 "The literature increasingly describes how various technical aids are used in lameness diagnostics [2]." I agree. However, it is a general statement. It would be more beneficial to tell the reader what various technical aids have been recently used in lameness diagnostics. See that the reference 2 is a short Letter/Commentary, not a systematic review.
This is now more detailed. Please see the rewritten passages.
L 33 Remove "According to previous studies". Everything you support with references is according to previous studies.
These sentences will be removed in the whole paper.
L 34-35 This is information that should be in the M&M section.
This was added to the M&M section and deleted here.
L 35 Remove "This has already been mentioned in previous publications". Everything you support with references is previously published.
This was removed.
L 37-38 "The recognition of lameness is an important point no matter in what way one deals with horses." What do you want to say with this sentence? Please remember that you are writing a scientific publication.
Okay. The whole paper underwent an English revision by an experienced veterinarian, who is a Native speaker. We hope these mistakes don’t occure anymore. This sentence was rewritten, because the whole introduction underwent a rewriting.
L 39 "naming lameness"? Do you mean clinically determining the degree and type of lameness, and limb affected by pain (of which lameness is a symptom) or identifying the cause of pain (diagnosis of lameness)? Support with appropriate reference.
We meant recognising the lameness (which limb, type and which degree). This sentence was rewritten, because the whole introduction underwent a rewriting.
L 42-44 See general comments and explain to me how you want to detect lameness in a standing position.
Pain leads to lameness. We chose horses with an unilateral hindlimb lameness detected by an lameness evaluation with the objective Equinosis Lameness Locator and an experienced veterinarian surgeon. After that, we knew which was the affected lame limb and we wanted to see, if the horses rest / unloads this leg more then the contralateral non affected, non-lame limb with the aid of accelerometers at stance. Plus how long the horses spent in a laying position compared to the non-lame group of horses.
L 47-50 This should not be part of the introduction section.
Okay. This was removed.
L 51 " detect lameness already at rest" I am sorry, but you cannot detect lameness (uneven gait) at rest.
Okay. We tried to remove this throughout the whole paper and make it more clear, what we actually mean.
L 52 You may assess limb loading/unloading and limb loading/unloading time, but this is not a detection of lameness.
Okay.
L 53-54 Be precise in what and how you quote, as references 10-12 do not describe the detection of lameness at rest.
Different references have been used for that. The whole paragraph underwent rewriting.
L 60 Remove generalization " which would be contrary to the hypothesis put forward."
Okay this was removed. We tried to remove all generalizations and be more precise.
L 68 You are examining the loading/unloading of a limb, which may or may not be related to pain in that limb, not lameness.
Okay. We tried to make it more clear.
L 70 What do you mean by "mild lameness"? What degree is it on the 0-5 AAEP scale? Which of these degrees do you think is objectively difficult to detect?
Okay we will be more precise about lameness degrees. We think grade 1 is most difficult to detect.
L 71 Since you claim to study horse behavior, what ethograms were assessed and how?
Hopefully with the new literature, we could make this point more clear and hope behaviour in horses is now mentioned more correctly.
L 74 How do you know that the posture behavior you are examining is due to pain? What pain? How severe? In what area?
We assumed that pain leads to lameness and picked the lame leg and wanted to compare the posture behaviour to a non-lame group of horses. The cause of lameness was not considered. But the horses had diagnostic anasthesia (= blocks), which were positiv in the lame horses. Which means, that the lameness was there due to pain. The measurements have taken place without blocks. Blocks have been done after that or have definetely worn off before. The severity of pain you will see in the lameness degrees and further on in the study.
L 76 The pain you assume is hypothetical. Have you performed diagnostic anesthesia on the affected limb to see if the symptoms/ posture behavior will stop or change after the pain is relieved?
The horses have been real clients horses at an equine hospital and underwent blocks, which relieved the pain and the horses haven’t been lame after that. The measurements haven’t been done while the block was still working. There is no comparison in the resting behaviour before and after the block or while the block is still working, but this could be a good study for the future and maybe even an useful tool.
L 81 Have you ruled out any other factors influencing whether the horse lies down or not?
The horses have been in a hospital environment, but it was quiet at night. No people around and no emergencies going on at night. Of course this new environment can influence the horses and the aim would be to monitor horses in their home environment. We just wanted to have a repeatable setting for the horses and showed that we have good statistical proven results. Now this could be transferred to an home environment.
L 89 What criteria? Describe clearly and legibly the criteria for including and excluding horses from the study.
The horses had to be unilateral hindlimb lame or non-lame. No other criteria.
L 93 The basis for determining lameness is a clinical examination, not the results of Lameness Locator measurements. Please describe the diagnostic procedure. Further on, you refer to the AAEP system, but you do not say anything about the study protocol.
We mentioned the lameness assessment above and it is added and written down in the M&Ms.
l 98-99 What do you mean by" In addition, the lameness must be recognizable to the Equinosis® Lameness Locator"?
It basically means, that the lameness Locator had to detect the lameness and also the experienced veterinarian. We added some more statistics from the lameness locator data to that.
L 100 Were the mares in heat or foal during the examination?
No.
L 102-106 Was this the horses' natural habitat? Were they moved to a homogeneous environment? Were the studies conducted in homogeneous conditions?
Every horse was used to being in a box. Yes it has been homogeneous.
The further part of the work can be assessed only after the assumptions, goal, hypothesis and the first part of the methodology related to the animal experimental system have been improved.
Okay.
As for prospective experimental studies, a lot of information that is important from the point of view of study design is missing, and the way of presenting what is collected is poor.
Okay.

Round 2
Reviewer 1 Report
Comments and Suggestions for Authors
The revised version of this manuscript is improved but further major revision is required before it could be considered potentially suitable for publication. The Methods remain incomplete/ inadequate to be able to repeat the study. Some of the results are presented in a confusing way. The Discussion is far too long relative to the scientific value of the study & lacks focus. The conclusions are exaggerated. The English remains inadequate. There are many grammatical and terminology errors.
Title non-lame would be preferable to sound horses
Abstract
Line 11 You said in the response the reviewer that the non-lame horses were at the clinic for other reasons so the term 'healthy horses' does not seem appropriate
line 18 significantly
Line 24 There should be no reference to figures in an abstract, which should be able to be read independently from the text
Of course the Vet Scores were different between groups – that was an inclusion criterion! Remove this.
Line 24 The Lameness Locator did not have asymmetry. Please rephrase. The English still requires major attention throughout the manuscript.
Line 27 less than rather than reduced
Line 28 more than rather than increased
Line 32 'By knowing the resting behaviour of an owners own horse, individual 32 changes in the loading and unloading of the hind limbs can be detected by the sensors and any 33 discomfort / pain and resulting lameness can be detected earlier, which will need further studies in 34 the future. '
This is a hypothesis - it is not a conclusion from this study - you have not studied horses in their home environment, nor have you followed horses longitudinally. Moreover, your study is biased towards horses with severe lameness and the results may be very different with mildly lame horses. In addition, your study only relates to hindlimbs. You continue to over-interpret the results of your study.
Line 36 'Thus, a reduction in downtime is achieved and the possibility of aggravation of a mild, objectively rather difficult to detect lameness is reduced. '
You have not proven this and this should not be part of an abstract - think about structuring an abstract as Background; Methods; Results; Limitations; Clinical relevance
Introduction
The Introduction has grown – it is too long – it should be more concise and focussed - providing reasons why the study was performed and the background behind your hypotheses.
Line 51 'An own pre study...' would be better phrased as a pilot study
Line 553 Non-lame - be consistent in terminology throughout – do not switch between healthy, sound and non-lame
Line 61-62 'Additionally, they may show 61 signs of stress and anxiety, including increased heart rate and cortisol levels ' There are a number of studies that have shown heart rate and cortisol concentrations are NOT good or reliable predictors of lameness’
Line 63 There is little current evidence to support the statement that horses with lameness lie down more. With chronic naturally occurring lameness horses may lie down less and experience sleep deprivation and be prone to episodic collapse as a result.
Line 68-9 This statement needs a reference.
Line 72 lying down - you lie down; a chicken lays an egg
Line 77 - 81 This contradicts what you have said previously; please be consistent
Write more succinctly
This could be summarised
In a study which compared horses with angular limb deformities (cases) with control horses, the case horses spent less time lying down.
The entire Introduction is long, repetitive, on places contradictory and needs to be more concise and accurate.
Line 89 AAEP is not defined
Please bear in mind that you current work, by the design of the study, cannot prove that you can make a diagnosis earlier - so please be realistic about what it can show
Line 92 'Accelerometers (MSR-145 mini data 92 logger) are used to detect the different resting behaviours mentioned above' This is methods
It was hypothesised that ...
Methods
How were the horses selected for inclusion in the study? Was it a random selection of non-lame horses and those with unilateral hindlimb lameness? Over what period of time was the study carried out?
Why were 7 horses assessed in 2017 & the remainder in 2020? That is a long interval. How might this have influence your results?
Line 112 Ten horses were not lame on the day of examination, had no recent history of lameness and had been in regular ridden work.
Line 117 All horses were assessed at walk and trot in straight lines on a firm surface
Line 117 which 3 gaits? How can you justify lungeing horses in canter with a grade 4 lameness?
Line 123 The table legend should be able to be read independently from the text.
Details of 20 horses with unilateral hindlimb lameness and 10 non-lame horses which underwent evaluation of resting behaviour using hindlimb accelerometers over 11 hours.
Line 126 - if a horse has been in the clinic for only up to 48 hours it will not be habituated to the environment. This of course makes it even more interesting that the lame horses were seen to lie down more than the non-lame horses. To me it is odd that the non-lame horses were observed to lie down so infrequently – having observed horses in a clinic for > 40 years this is very unusual.
Line 135 'It was important not to disregard the pre-programmed assignment of 135 the sensors to the legs. ' What does this mean?
Line 145 Companion infers that this was a horse known to it which I suspect was not the case.
Every horse had a horse in an adjacent box or across the aisle
Line 164 'As has already been described, accelerometers are good at recognizing different behaviors. ' This is explained in detail from line 170 - avoid repetition
Line 169 'As has already been described, accelerometers are good at recognizing different behaviors. ' This is redundant
Avoid replication.
The more concise descriptions are the better
Line 178 & elsewhere - please use consistent terminology - either hindlimb or hind leg or pelvic limb - do not switch between
Please make it clear that only hindlimb data were used in this study
Line 182 a horse, not the horse
Applies elsewhere as well
Line 195 the data were
Line 198 the Lameness Locator may identify asymmetry - it does not have asymmetry
Results
Line 206 What do you mean by mean of the lameness locator?
Remember that table legends need to be readable in isolation from the text so you need to refer to 20 horses with unilateral HL lameness and 10 control non-lame horses - & somewhere you are going to have to discuss that some of your none-lame horses had asymmetry values outside the so-called normal range - & greater than some of your lame horses. In addition some of your lame horses had values less than the so-called threshold value for lameness - this is a can of worms!
Line 207 non-lame link - I presume that you mean non-lame limb.
Laytime is not acceptable - time spent lying down - this would be more comprehensible as minutes not seconds!
You need to define in the table legend what is meant by resting relation
For horses 8,9,10,12,15, 17, 19, 22, 24, 28 under AAEP lameness score what does 2,5; 1,5; 3,5 etc. mean? If you are using the AAEP scale you can only have integers; if you are modifying the AAEP scale then you have to inform the reader how you are modifying it. The methods must be reproducible by a reader.
How do you explain the apparent disparity between the grades assigned and the asymmetry determined by the Lameness Locator, for example Horse 10?
Table 3 - see comments re table 2
Line 213 please present all results in the past tense
Line 216 less lame than what?
What is below? - this is not illustrated in Fig. 2
Line 218 See previous comments about figure and table legends
Line 221 ALL results should be in the past tense
Lying down
Lied down
Lame horses spent significantly greater time lying down than non-lame horses.
‘…and supports the third hypothesis’ This does not belong in Results
Line 225 see previous comments about figure & table legends. Lay time is not appropriate terminology. The time spent lying down.
You need to explain the outliers in the figure legend
I suggest that the time spent lying down is expressed as minutes rather than seconds
Line 228 'which you can see in' is redundant. Figure 4 is completely unnecessary because by definition the control horses were non-lame and so score 0. This statistic is also redundant.
Please be consistent in terminology - stick to non-lame rather than healthy here & elsewhere
Line 229 The Lameness Locator does not have asymmetry
Objectively measured asymmetry
Line 233 'while the lying time was significantly in-232 creased (p= 0.013) in the lame horses (5040 ± 4268) compared to the healthy horses (996 ± 233 2108)' - this does not seem to fit with the results presented in Fig. 3 for the non-lame horses
Line 235 Laytime is not an appropriate term
Line 237'which you can see in ' is redundant
Line 239 Line See previous comments about figure & table legends - this legend is completely inadequate. It must be made clear that the data reflect objectively measured asymmetry in mm.
The overlap between the 2 groups is of some cause for concern ....
Lines 240 - 246 This does not belong in Results.
Remove
Discussion
The Discussion is FAR too long relative to the scientific content of the paper and lacks focus.
I suggest that it is completely rewritten bearing in mind that the purpose of a Discussion is to concisely & briefly:
1. Summarise your results relative to your hypotheses
2. Discuss your results relative to current published knowledge
3. Discuss any anomalies that have arisen in your results
4. Summarise the limitations of your study
5. Suggest what needs to be done next e.g., determine the minimum time for which horses need to be monitored; observe horses in their home environment; observe horses during day time which is when horse carers are likely to be about
6. Briefly summarise your conclusions
Do not make exaggerated claims - you have no evidence to indicate that you can detect lameness earlier than by regular monitoring of gait in hand, on the lunge & ridden & observations about ridden horse behaviour. Avoid saying that this is a tool to improve equine welfare until you have more evidence. At the moment it would be completely impractical for owners to be using this tool in the field. You must be realistic. However it probably is reasonable to say that a horse which persistently rests a single hindlimb - and the same hindlimb on a highly repeatable basis probably has a problem.
Make sure that you accurately cite the results of other studies - you do not always do so.
Bear in mind the lameness severity in your horses - the results would not necessarily translate, for example, to horses that only showed hindlimb lameness when ridden and did not show visible or measurable asymmetry outside the reference range in hand.
Do not get side tracked into discussing anything that is not of direct relevance to your study.
I will not make more detailed comments because I think that this Discussion needs radical change.
Appendices
Line 572 Be consistent in the use of terminology - hindlimb
Throughout this appendix be consistent - e.g., non-lame not sound
Comments on the Quality of English Language
Needs a lot of work
Author Response
Please see as well the attached document.
The revised version of this manuscript is improved but further major revision is required before it could be considered potentially suitable for publication. The Methods remain incomplete/ inadequate to be able to repeat the study. Some of the results are presented in a confusing way. The Discussion is far too long relative to the scientific value of the study & lacks focus. The conclusions are exaggerated. The English remains inadequate. There are many grammatical and terminology errors.
Please see all the comments below and the new version of the manuscript. All new changes are made in blue now to see the difference immediately.
Title non-lame would be preferable to sound horses
Please see the changed title to: Monitoring of Non-lame Horses and Horses with Unilateral Hindlimb Lameness at Rest with the Aid of Accelerometers.
Abstract
Line 11 You said in the response the reviewer that the non-lame horses were at the clinic for other reasons so the term ‚healthy horses‘ does not seem appropriate
Yes you are right, sorry. I changed this to non-lame.
Line 18 significantly
This is corrected.
Line 24 There should be no reference to figures in an abstract, which should be able to be read independently from the text
Okay, sorry please see the newly constructed abstract and I removed the references to figures.
Of course the Vet Scores were different between groups – that was an inclusion criterion! Remove this.
This was removed and the whole Abstract was shortened and rewritten.
Line 24 The Lameness Locator did not have asymmetry. Please rephrase. The English still requires major attention throughout the manuscript.
The asymmetry was removed. The English was also checked again throughout the whole manuscript.
Line 27 less than rather than reduced
Reduced was changed to less.
Line 28 more than rather than increased
Increased was changed to more.
Line 32 ‚By knowing the resting behaviour of an owners own horse, individual 32 changes in the loading and unloading of the hind limbs can be detected by the sensors and any 33 discomfort / pain and resulting lameness can be detected earlier, which will need further studies in 34 the future. ‚
This is a hypothesis – it is not a conclusion from this study - you have not studied horses in their home environment, nor have you followed horses longitudinally. Moreover, your study is biased towards horses with severe lameness and the results may be very different with mildly lame horses. In addition, your study only relates to hindlimbs. You continue to over-interpret the results of your study.
Was changed into: Future studies will show, if the knowledge of the individual resting pattern will help to detect any discomfort/pain at an early stage.
Line 36 ‚Thus, a reduction in downtime is achieved and the possibility of aggravation of a mild, objectively rather difficult to detect lameness is reduced.
You have not proven this and this should not be part of an abstract – think about structuring an abstract as Background; Methods; Results; Limitations; Clinical relevance
Please see the new abstract, which was shortened.
Introduction
The Introduction has grown – it is too long – it should be more concise and focussed - providing reasons why the study was performed and the background behind your hypotheses.
Please see the shortened introduction.
Line 51 ‚An own pre study…‘ would be better phrased as a pilot study
This was changed to pilot study
Line 553 Non-lame - be consistent in terminology throughout – do not switch between healthy, sound and non-lame
Every synonym was changed to non-lame.
Line 61-62 ‚Additionally, they may show 61 signs of stress and anxiety, including increased heart rate and cortisol levels ‚ There are a number of studies that have shown heart rate and cortisol concentrations are NOT good or reliable predictors of lameness’
This sentence was removed and the introduction was shortened in total.
Line 63 There is little current evidence to support the statement that horses with lameness lie down more. With chronic naturally occurring lameness horses may lie down less and experience sleep deprivation and be prone to episodic collapse as a result.
Please see the evidence of this study, which was included in the text now. A lot of this might be answered by the evidences in the discussion.
Line 68-9 This statement needs a reference.
A reference was added.
Line 72 lying down – you lie down; a chicken lays an egg
This was changed throughout the manuscript.
Line 77 – 81 This contradicts what you have said previously; please be consistent
Write more succinctly
This could be summarised
In a study which compared horses with angular limb deformities (cases) with control horses, the case horses spent less time lying down.
Thank you for the suggestion. This sentence was used to replace the actual sentence.
The entire Introduction is long, repetitive, on places contradictory and needs to be more concise and accurate.
The introduction is shortened and hopefully more concise and accurate now.
Line 89 AAEP is not defined
The definition was added.
Please bear in mind that you current work, by the design of the study, cannot prove that you can make a diagnosis earlier – so please be realistic about what it can show
The abstract was changed like this: Future studies will show, if the knowledge of the individual resting pattern will help to detect any discomfort/pain at an early stage.
Line 92 ‚Accelerometers (MSR-145 mini data 92 logger) are used to detect the different resting behaviours mentioned above‘ This is methods
This was removed here and added to the methods.
It was hypothesised that …
This was changed.
Methods
How were the horses selected for inclusion in the study? Was it a random selection of non-lame horses and those with unilateral hindlimb lameness? Over what period of time was the study carried out?
Yes, the horses were selected by the case load in the hospital. As soon as a horse was suitable (either non-lame or unilateral hindlimb lame), it was used as long as it was ok for the owners to participate in the study. That is the reason why the data collection needed so much time. It was a small hospital with an ambulatory practice, so not many horses were brought to the hospital for a lameness evaluation, because a lot of them have been solved on the road. The main study was carried out over a year. After that year, few horses were added from the pilot study in 2017, which were absolutely suitable for this study as well.
Why were 7 horses assessed in 2017 & the remainder in 2020? That is a long interval. How might this have influence your results?
The main study was carried out over a year. After that year, few horses were added from the pilot study in 2017, which were absolutely suitable for this study as well. This might not have influenced the results, because the data was the same and the horses suitable. I have got Ethics and client consent for this. I added the ethic sign in the paper.
Line 112 Ten horses were not lame on the day of examination, had no recent history of lameness and had been in regular ridden work.
Thank you for the suggestion, this sentence was added / changed.
Line 117 All horses were assessed at walk and trot in straight lines on a firm surface
Thank you for the suggestion, this sentence was added / changed.
Line 117 which 3 gaits? How can you justify lungeing horses in canter with a grade 4 lameness?
Thank you for your suggestion. I have clarified that in horses with grade 4 lameness, the canter was not always possible.
Line 123 The table legend should be able to be read independently from the text.
Please see all the new legends. Resting relation is defined as non-lame resting time / lame resting time within the lame group of horses and right limb resting time / left limb resting time within the non-lame group of horses. This was added in the text (results).
Details of 20 horses with unilateral hindlimb lameness and 10 non-lame horses which underwent evaluation of resting behaviour using hindlimb accelerometers over 11 hours.
Thank you for the suggestion, this was used as a legend for Table 1.
Line 126 – if a horse has been in the clinic for only up to 48 hours it will not be habituated to the environment. This of course makes it even more interesting that the lame horses were seen to lie down more than the non-lame horses. To me it is odd that the non-lame horses were observed to lie down so infrequently – having observed horses in a clinic for > 40 years this is very unusual.
Yes, you are right, this is very unusual. But it was a significant finding. We can only guess why it is like that. Maybe the non-lame horses were influenced by the unfamiliar environment, whereas the lame horses „needed“ to lie down under any circumstances because of pain. This would support our third hypothesis and so the time, which they spent lying down can also be an indicator for horses with a mild lameness of 1 or 2/5 degree.
Line 135 ‚It was important not to disregard the pre-programmed assignment of 135 the sensors to the legs. ‚ What does this mean?
The sensors were programmed for each leg. It was important to use right hind for right hind and left hind for left hind. Also up and down diraction always had to be the same.
Line 145 Companion infers that this was a horse known to it which I suspect was not the case.
Sometimes it was the case, that owners brought two horses. One patient and one companion, because they couldn’t keep one horse alone at home or the horses were „friends“ and couldn’t be seperated. Does that answer you question?
Every horse had a horse in an adjacent box or across the aisle
The word adjacent was added.
Line 164 ‚As has already been described, accelerometers are good at recognizing different behaviors. ‚ This is explained in detail from line 170 – avoid repetition
We hope all repetitions are deleted now.
Line 169 ‚As has already been described, accelerometers are good at recognizing different behaviors. ‚ This is redundant
Avoid replication.
The more concise descriptions are the better
This was removed.
Line 178 & elsewhere - please use consistent terminology – either hindlimb or hind leg or pelvic limb – do not switch between
The term used throughout the whole manuscript is limb / hindlimbs now.
Please make it clear that only hindlimb data were used in this study
We hope this is more clear now. You will find it in the methods.
Line 182 a horse, not the horse
Applies elsewhere as well
This was changed througout the whole manuscript.
Line 195 the data were
This was changed.
Line 198 the Lameness Locator may identify asymmetry – it does not have asymmetry
This was changed.
Results
Line 206 What do you mean by mean of the lameness locator?
The values, which were generated by the lameness locator were meant. It creates a minimum and maximum difference in mm. We combined the HDmin and HDmax of the Lameness Locator. We changed it also to the asymmetry evaluated by the lameness locator sometimes in the text.
Remember that table legends need to be readable in isolation from the text so you need to refer to 20 horses with unilateral HL lameness and 10 control non-lame horses - & somewhere you are going to have to discuss that some of your none-lame horses had asymmetry values outside the so-called normal range - & greater than some of your lame horses. In addition some of your lame horses had values less than the so-called threshold value for lameness – this is a can of worms!
Please see all the new legends.
Line 207 non-lame link – I presume that you mean non-lame limb.
Yes sorry, that was changed.
Laytime is not acceptable - time spent lying down – this would be more comprehensible as minutes not seconds !
Laytime was removed throughout the whole manuscript.
Also seconds were changed to minutes.
You need to define in the table legend what is meant by resting relation
Resting relation is defined as non-lame resting time / lame resting time within the lame group of horses and right limb resting time / left limb resting time within the non-lame group of horses. This was added in the text (results).
For horses 8,9,10,12,15, 17, 19, 22, 24, 28 under AAEP lameness score what does 2,5; 1,5; 3,5 etc. mean? If you are using the AAEP scale you can only have integers; if you are modifying the AAEP scale then you have to inform the reader how you are modifying it. The methods must be reproducible by a reader.
We added a study where half digits where used as well. It was defined because of the presence of multiple veterinarians. Hopefully this is stated clearer now.
How do you explain the apparent disparity between the grades assigned and the asymmetry determined by the Lameness Locator, for example Horse 10?
To be honest, we don’t know, but there are studies, which conclude, that for horses with mild lameness subjective evaluation of lameness is not very reliable, because there can be differences between humans and instrumented lameness evaluation systems. Maybe this is the case here.
- VetRec., 2019 Jan 12;184(2):63. doi: 10.1136/vr.105058. Reliability of equine visual lameness classification as a function of expertise, lameness severity and rater confidence
Sandra Dorothee Starke , Maarten Oosterlinck
In the abstract: Visual equine lameness assessment is often unreliable, yet the full understanding of this issue is missing (Starke and Oosterlinck, 2019).
- Equine Vet J 2010 Mar;42(2):92-7. doi: 10.2746/042516409X479568. Repeatability of subjective evaluation of lameness in horses
K G Keegan 1 , E V Dent, D A Wilson, J Janicek, J Kramer, A Lacarrubba, D M Walsh, M W Cassells, T M Esther, P Schiltz, K E Frees, C L Wilhite, J M Clark, C C Pollitt, R Shaw, T Norris
Table 3 – see comments re table 2
Please see the changed legends.
Line 213 please present all results in the past tense
All results are presented in the past tense now.
Line 216 less lame than what?
Horses with a lameness degree of maximal 1-2/5 degree.
What is below? – this is not illustrated in Fig. 2
The asymmetry evaluated by the lameness locator (ALL) showed a significant difference (p= 0.025) between the lame group (10.5 ± 5.6 mm) and the healthy group (5.8 ± 3.4 mm), which you can see in Figure 5. This was added to the text.
Line 218 See previous comments about figure and table legends
Yes. We hope it is now more clear.
Line 221 ALL results should be in the past tense
All results are presented in the past tense now.
Lying down
Ok.
Lied down
Ok.
Lame horses spent significantly greater time lying down than non-lame horses.
This sentence was used.
‘…and supports the third hypothesis’ This does not belong in Results
This was removed.
Line 225 see previous comments about figure & table legends. Lay time is not appropriate terminology. The time spent lying down.
This was changed throughout the whole document.
You need to explain the outliers in the figure legend
Figure 2: Horse 6, the 5th lame horse, is marked as an outlier in the first two boxplots (circle and number 5).
Figure 3: Horses 7 and 14, which were the 2nd and 3rd non-lame horses, are marked as an outlier in the right boxplot (circle and number 2 and 3).
I suggest that the time spent lying down is expressed as minutes rather than seconds
This was changed to minuets.
Line 228 ‚which you can see in‘ is redundant. Figure 4 is completely unnecessary because by definition the control horses were non-lame and so score 0. This statistic is also redundant.
This was removed.
Please be consistent in terminology - stick to non-lame rather than healthy here & elsewhere
This was changed to non-lame throughout the whole manuscript.
Line 229 The Lameness Locator does not have asymmetry
Objectively measured asymmetry
Asymmetry identified by the lameness locator was used.
Line 233 ‚while the lying time was significantly in-232 creased (p= 0.013) in the lame horses (5040 ± 4268) compared to the healthy horses (996 ± 233 2108)‘ - this does not seem to fit with the results presented in Fig. 3 for the non-lame horses
We have now changed the scale to minutes (see below). The boxplots in Figure 3 show the median (line in the box) and the quartiles. Mean±SD is given in the text. Especially in the right boxplot is a large deviation between the median and the mean due to the outliers. The outliers are not taken into account for the boxplot, so the median and quartiles are zero.
Line 235 Laytime is not an appropriate term
This was changed to time spent lying down in the whole manuscript.
Line 237’which you can see in ‚ is redundant
This was deleted.
Line 239 Line See previous comments about figure & table legends - this legend is completely inadequate. It must be made clear that the data reflect objectively measured asymmetry in mm.
This was changed.
The overlap between the 2 groups is of some cause for concern ….
(Figure 4)
Yes, of course the boxplots do not show a clear distinction line between the groups, but the statistic comparison showed a significant difference. See: The asymmetry identified by the lameness locator (ALL) showed a significant difference (p= 0.025) between the lame group (10.5 ± 5.6 mm) and the non-lame group (5.8 ± 3.4 mm).
Lines 240 – 246 This does not belong in Results.
Remove
This was removed.
Discussion
The Discussion is FAR too long relative to the scientific content of the paper and lacks focus.
I suggest that it is completely rewritten bearing in mind that the purpose of a Discussion is to concisely & briefly:
- Summarise your results relative to your hypotheses
- Discuss your results relative to current published knowledge
- Discuss any anomalies that have arisen in your results
- Summarise the limitations of your study
- Suggest what needs to be done next e.g., determine the minimum time for which horses need to be monitored; observe horses in their home environment; observe horses during day time which is when horse carers are likely to be about
- Briefly summarise your conclusions
Do not make exaggerated claims – you have no evidence to indicate that you can detect lameness earlier than by regular monitoring of gait in hand, on the lunge & ridden & observations about ridden horse behaviour. Avoid saying that this is a tool to improve equine welfare until you have more evidence. At the moment it would be completely impractical for owners to be using this tool in the field. You must be realistic. However it probably is reasonable to say that a horse which persistently rests a single hindlimb – and the same hindlimb on a highly repeatable basis probably has a problem.
Make sure that you accurately cite the results of other studies – you do not always do so.
Bear in mind the lameness severity in your horses – the results would not necessarily translate, for example, to horses that only showed hindlimb lameness when ridden and did not show visible or measurable asymmetry outside the reference range in hand.
Do not get side tracked into discussing anything that is not of direct relevance to your study.
I will not make more detailed comments because I think that this Discussion needs radical change.
Due to the major changes in the discussion, we can’t comment on everything in detail here. Please see the new text in the manuscript. Thank you.
Appendices
Line 572 Be consistent in the use of terminology – hindlimb
This was changed throughout the whole appendix.
Throughout this appendix be consistent – e.g., non-lame not sound
This was changed throughout the whole appendix.
Comments on the Quality of English Language
Needs a lot of work
The language has been proved again, which hopefully makes the whole manuscript more comprehensable and shorter.

Reviewer 2 Report
Comments and Suggestions for Authors
Dear authors,
I appreciate the effort put into improving this manuscript especially with regard to the diagnosis of lameness. We still do not know the cause of lameness, which was not raised as a limitation in the discussion. We also do not know the ethogram of behavior assessment, since the authors still maintain that the time of "resting behavior" is a type of behavior. It would be much clearer if the authors discussed exactly what they are measuring - time of loading and time of unloading, because behavior itself consists of many more variables than just the results of accelerometer measurements.
The citation in L 53 still needs improvement.
The discussion after the changes introduced became very long and difficult to follow, but there is visible progress.
Author Response
Dear authors,
I appreciate the effort put into improving this manuscript especially with regard to the diagnosis of lameness. We still do not know the cause of lameness, which was not raised as a limitation in the discussion. We also do not know the ethogram of behavior assessment, since the authors still maintain that the time of "resting behavior" is a type of behavior. It would be much clearer if the authors discussed exactly what they are measuring - time of loading and time of unloading, because behavior itself consists of many more variables than just the results of accelerometer measurements.
All new changes have been made in the color blue. That we didn’t include the cause of lameness is mentioned in the discussion. The word behavior was changed throughout the whole manuscript to either “(un)loading limbs” (where possible in the context) or “resting pattern”.
The citation in L 53 still needs improvement.
This was changed to: Based on a pilot study (1), this study compared the resting pattern of non-lame and lame horses. Non-lame horses display a wide range of behaviors, including exercise, feeding, and social interactions.
Please also see the citation changed to the required style.
The discussion after the changes introduced became very long and difficult to follow, but there is visible progress.
Please see the rewritten discussion part.
Round 3
Reviewer 1 Report
Comments and Suggestions for Authors
The second revision of this paper is much improved but in my opinion it is still not acceptable for publication in its current form. The Discussion is far too long relative to the scientific value of the study & lacks focus. The conclusions are exaggerated.
In line 34 you imply that objective gait analysis is the most accurate method of lameness diagnosis - but this is dispelled by your data. You are going to have to define what you mean by non-lame in the context of this study. Some of your non-lame horses had asymmetry values outside the so-called normal reference range and larger than some of your lame horses.
Line 64 How is regularly defined? How was this determined? Are any results presented that support this?
There needs to be some discussion about the discrepancies between the lameness grades and the measured asymmetries, for example for horse 5.
You also need to justify the inclusion of horses with measurable asymmetry outside the so called normal range as non-lame horses
I think that you also need to discuss why only 2 of your non-lame horses were seen to lie down for any significant length of time - why might that be? - obviously they were at the clinic because they had a clinical problem - was it this that influenced their lack of resting behaviour during the night? This is what is abnormal – rather than the lame horses lying down more. To me it is odd that the non-lame horses were observed to lie down so infrequently – having observed horses in several clinics for > 40 years this is very unusual.
Do not make exaggerated claims - you have no evidence to indicate that you can detect lameness earlier by observing resting patterns than by regular monitoring of gait in hand, on the lunge & ridden & observations about ridden horse behaviour. Avoid saying that this is a tool to improve equine welfare until you have more evidence. At the moment it would be completely impractical for owners to be using this tool in the field. You must be realistic. However it probably is reasonable to say that a horse which persistently rests a single hindlimb - and the same hindlimb on a highly repeatable basis - has a problem.
Make sure that you accurately cite the results of other studies - you do not always do so.
Bear in mind the lameness severity in your horses - the results would not necessarily translate, for example, to horses that only showed hindlimb lameness when ridden and did not show visible or measurable asymmetry outside the reference range in hand.
You have no idea when asymmetrical resting of a hindlimb may develop relative to the onset of lameness – so you cannot say that the technique will permit early diagnosis of lameness.
Although you have statistically significant results you have large overlap between groups, so avoid over-interpretation of your results. Statistics can be misleading and should not be misused.
It is highly improbable that owners are going to use accelerometers on a regular basis – but to encourage observation of resting patterns would be potentially beneficial – especially if the patterns were also observed during the normal hours during which horses are observed.
There are some incorrect reference citations.
More specific comments:
Line 10 lame limb (for consistency)
Line 15 lying down
Line 17 The results presented in the Abstract should all be in the past tense
Lines 20-21 Please rephrase this - I am not clear what you are saying here
'The group of lame horses (13 ± 11%) lie down significantly longer (p = 0.012) than non-lame horses (3 ± 6%).’
Line 26 how individual resting patterns may help to detect pain
Line 48 'In addition, lame horses spend more time lying down' - a reference is required to support this statement
Line 51 10. Weishaupt, M.A.; Wiestner, T.; Hogg, H.P.; Jordan, P.; Auer, J.A. Compensatory load redistribution of horses with induced weight-bearing forelimb lameness trotting on a treadmill. Vet. J. 2006, 171, 135–146, 498 doi:10.1016/j.tvjl.2004.09.004.
How does a paper describing experimentally induced forelimb lameness support the statement 'These differences may be attributed to the pain and discomfort associated with lameness and therefore provide valuable insights into a horse's well-being .'?
Line 60 were
Line 77 The data were collected over one year.
You say some suitable horses were added from the pilot study - how was 'suitable’ defined?
Or say 'In addition horses examined during the pilot study which fulfilled the inclusion criteria were also included.'
Line 82 Twenty (numbers should be written in full at the beginning of a sentence)
Line 82 How can you say with certainty that the lameness was unilateral unless lameness was abolished by diagnostic anaesthesia and did not switch to the contralateral limb?
If this was not done then this should be mentioned as a limitation of your study.
You imply in line 96 that diagnostic anaesthesia was used. This would be better mentioned here.
e.g., All lame horses underwent diagnostic anaesthesia of the lame lime and in none did lameness switch to the contralateral limb.
Line 90 11 is a cattle reference not a horse reference! I don't think this is particularly appropriate.
This sentence does not read easily - please rephrase.
Line 95 when (not where)
Lines 96-98 - see previous comment
Line 102 AAEP scale
I think that you need to explain in the figure legend what is meant by, for example, 2-3/5, because this is not consistent with what you have said in the test when you said you used halves and used the mean values for the two observers. So in this Table if you say 2-3 do you mean that one observer graded the lameness as 2 and the other as 3? There is lack of consistency in presentation in Tables 1 & 2.
Line 104 approximately
Line 116 The method would be better than the way
Line 149 were performed
Line 152 minutes (not minuets!)
Line 159 were analysed
Line 166 were calculated
Line 172 are summarised
In the legend for both Tables 1 & 2 the term 'Resting relation' needs to be defined. Remember the legend must be able to be read independently from the text
Table 2 we need to know what is meant by the asymmetry measurement given of the Lameness Locator - was this MinDiff, MaxDiff?
It is rather confusing that the Vet Score is different to that in Table 1 because Table 1 gives a range whereas as Table 2 gives halves
Line 183 This needs to be expressed differently
The lame limb was loaded for less time than the non-lame limb (p=0.035).
Line 185-6'The total of the lame horses (13 ± 185 11%) lied down significantly longer (p= 0.01) than the group of non-lame horses (3 ± 6%) ' Please rephrase this - I don't know what you are saying here
Line 187 - 9 ‘Resting relation was defined as non-lame resting time / lame resting time within the lame group of horses and right limb resting time / left limb resting time within the non-lame group of horses. ‘
This repeats the M&M and doesn't belong here.
Figure 2 The large SD and the overlap between the two groups of horses needs to be discussed.
Legend for Fig. 2 Relative resting time needs to be defined in the legend
I think that you misunderstood my comments about the outlier. The reason for this outlier needs to be included in the Discussion. The number of the horse in the figure should marry up with the horse number in the tables - it is otherwise confusing even if explained in the legend - change the figure if this is actually horse 6
Line 195 All Results should be written in the past tense
Line 195 what is meant by regularly in this context?
Lines 196 and 199 - you don't need to refer back to the hypotheses in the results - you should merely state the results
Figure 3 Please amend your figure so that the outliers are the correct horse numbers - do not create unnecessary confusion - even if you attempt to explain this in the legend. Keep things consistent and simple.
Line 209'(5040 ± 4268) 209 compared to the non-lame horses (996 ± 2108). What units are these numbers?
Line 214 The asymmetry did not show;
the mean asymmetry was different between the lame and non-lame horses.
Here and elsewhere it is unnecessary to say significant or significantly if the p value is <0.05.
Figure 4 20 horses with unilateral hindlimb lameness and 10 non-lame horses
This Discussion remains FAR TOO LONG - it must be shortened and made much more focussed and concise as indicated previously
You need to justify how you can call non-lame horses which have asymmetry values outside the so-called normal range & sometimes higher than those observed in the lame horses
You need to think about why the so-called non-lame horses were lying down less than the lame horses - this is abnormal - & needs to be thought of like that
Lines 224 -7 It is unnecessary to repeat the hypotheses
You could say
In accordance with our hypotheses non-lame horses rested their hindlimbs equally etc.
Line 227 I think that you have to consider the duration of time for which the lame horses had been lame - how might that influence your results?
Line 228 - 242 This is all conjecture and should be expressed much more concisely.
Line 247 'How this influences the actual resting patterns would be a good investment for future studies. ' I don't think investment is an appropriate word in this context
We know that horses with severe lameness may 'tread' (see Torcivia for definition)
Line 250-1 However, this would also be a consideration that can be taken into account in future experiments on this topic.
You did not do an experiment! I think that you mean that these factors merit further investigation
Line 252 - 263 - much of this is irrelevant and /or speculation
All you need to say that variation among breeds in your study may have influenced the results
Reference 30 is a review article & I do not believe that it provides any evidence about specific breed differences
Line 259 I don't think that reference 32 is applicable here
Lines 263-267 This has to be put into the context that we know that chronic orthopaedic pain can be a reason for failure to lie down, sleep deprivation and a tendency to collapse
Line 270 smaller than what?
Line 276 'Whether the time of signaling in horses increases with a higher degree of lameness would have to be specifically investigated in a new experiment.’ What does this mean?
Line 284 -88 This could be expressed much more concisely
Lines 289 - 295 This is not really relevant - what you need to discuss is the overlap between measured asymmetry between the lame horses and the non-lame horses and the relationship between what you measured and the published threshold values for lameness
Line 307 led , not lead
Lines 310 - 14 This is not really necessary
Line 332 - 336 this belongs earlier
Line 336 - As I have said before I think it is more remarkable that your non-lame horses were so abnormal in their lying down patterns. Why?
Lines 344 - 356 Huge amount of repetition here
This Discussion remains unfocussed and there are long rambling sections that are not related directly to the results of your study - you must relate your comments back to your results
Avoid repeating what you've already said in the discussion in the conclusions
Comments on the Quality of English Language
Still not good enough, although much improved.
Author Response
Please see as well the attached document.
Open Review
(x) I would not like to sign my review report
( ) I would like to sign my review report
Quality of English Language
( ) I am not qualified to assess the quality of English in this paper.
( ) The English is very difficult to understand/incomprehensible.
( ) Extensive editing of English language required.
( ) Moderate editing of English language required.
(x) Minor editing of English language required.
( ) English language fine. No issues detected.
YesCan be improvedMust be improvedNot applicableDoes the introduction provide sufficient background and include all relevant references?( )( )(x)( )Is the research design appropriate?( )( )(x)( )Are the methods adequately described?( )( )(x)( )Are the results clearly presented?( )( )(x)( )Are the conclusions supported by the results?( )( )(x)( )
Comments and Suggestions for Authors
The second revision of this paper is much improved but in my opinion it is still not acceptable for publication in its current form. The Discussion is far too long relative to the scientific value of the study & lacks focus. The conclusions are exaggerated.
All new changes are done in purple. Please see the rewritten Discussion. To shorten it, we removed some parts, which we thought were wished after the first round. And please also see the new conclusions.
In line 34 you imply that objective gait analysis is the most accurate method of lameness diagnosis - but this is dispelled by your data. You are going to have to define what you mean by non-lame in the context of this study. Some of your non-lame horses had asymmetry values outside the so-called normal reference range and larger than some of your lame horses.
Objective gait analysis can be more accurate then subjective by a vet. The two vets in this study sometimes also disagreed. Non-lame is defined by the absence of lameness on that day of examination.
Line 64 How is regularly defined? How was this determined? Are any results presented that support this?
Regularly means that the resting "events" took place in the same horse in the same way. No results present this. It was deleted.
There needs to be some discussion about the discrepancies between the lameness grades and the measured asymmetries, for example for horse 5.
Please see the discussion about this.
You also need to justify the inclusion of horses with measurable asymmetry outside the so called normal range as non-lame horses
The horses were included based on the Vet Score. A perfect match between Vet Score and ALL is very unlikely. However, the statistical comparison showed a significant correlation between Vet Score and ALL.
I think that you also need to discuss why only 2 of your non-lame horses were seen to lie down for any significant length of time - why might that be? - obviously they were at the clinic because they had a clinical problem - was it this that influenced their lack of resting behaviour during the night? This is what is abnormal – rather than the lame horses lying down more. To me it is odd that the non-lame horses were observed to lie down so infrequently – having observed horses in several clinics for > 40 years this is very unusual.
Measurements for this study were taken at night to capture natural resting patterns with minimal disturbance, although the clinic setting may have influenced the results due to the unfamiliar environment. Our results may reflect that the horses are in an unfamiliar environment, especially the group of non-lame horses that do not lie down. Uncertainty in the data can certainly be reduced if the horses are analyzed in their usual environment.
Do not make exaggerated claims - you have no evidence to indicate that you can detect lameness earlier by observing resting patterns than by regular monitoring of gait in hand, on the lunge & ridden & observations about ridden horse behaviour. Avoid saying that this is a tool to improve equine welfare until you have more evidence. At the moment it would be completely impractical for owners to be using this tool in the field. You must be realistic. However it probably is reasonable to say that a horse which persistently rests a single hindlimb - and the same hindlimb on a highly repeatable basis - has a problem.
Please see the new explanations. We deleted the interpretation about the well- being of a horse. Also we hopefully now don't make any suggestions about early diagnosis of lameness anymore.
Make sure that you accurately cite the results of other studies - you do not always do so.
Okay. We tried to fix this.
Bear in mind the lameness severity in your horses - the results would not necessarily translate, for example, to horses that only showed hindlimb lameness when ridden and did not show visible or measurable asymmetry outside the reference range in hand.
Please see the changed text: Due to the lameness severity in the participating horses the results would not necessarily translate, for example to horses that only showed hindlimb lameness when ridden and did not show visible or measurable asymmetry outside the reference range in hand.
You have no idea when asymmetrical resting of a hindlimb may develop relative to the onset of lameness – so you cannot say that the technique will permit early diagnosis of lameness.
We hope that we now deleted all parts, which still state this.
Although you have statistically significant results you have large overlap between groups, so avoid over-interpretation of your results. Statistics can be misleading and should not be misused.
The limitations of the study were pointed out in the discussion.
It is highly improbable that owners are going to use accelerometers on a regular basis – but to encourage observation of resting patterns would be potentially beneficial – especially if the patterns were also observed during the normal hours during which horses are observed.
This was added to the discussion like this: Using accelerometers in a daily basis would be a good tool, but might be impractical for owners. But the observation of resting patterns, also during the day, can be useful.
There are some incorrect reference citations.
Okay we tried to fix that.
More specific comments:
Line 10 lame limb (for consistency)
This was changed.
Line 15 lying down
This was changed.
Line 17 The results presented in the Abstract should all be in the past tense
All of this was changed to the past tense.
Lines 20-21 Please rephrase this - I am not clear what you are saying here
'The group of lame horses (13 ± 11%) lie down significantly longer (p = 0.012) than non-lame horses (3 ± 6%).’
Christian
This was rephrased to: The lame horses (13 ± 11%) lie down significantly longer (p = 0.012) than non-lame horses (3 ± 6%).
Line 26 how individual resting patterns may help to detect pain
This was changed.
Line 48 'In addition, lame horses spend more time lying down' - a reference is required to support this statement
This was added.
Line 51 10. Weishaupt, M.A.; Wiestner, T.; Hogg, H.P.; Jordan, P.; Auer, J.A. Compensatory load redistribution of horses with induced weight-bearing forelimb lameness trotting on a treadmill. Vet. J. 2006, 171, 135–146, 498 doi:10.1016/j.tvjl.2004.09.004.
How does a paper describing experimentally induced forelimb lameness support the statement 'These differences may be attributed to the pain and discomfort associated with lameness and therefore provide valuable insights into a horse's well-being .'?
All sentences, which are about a horses well-being were removed.
Line 60 were
This was changed.
Line 77 The data were collected over one year.
This was changed.
You say some suitable horses were added from the pilot study - how was 'suitable’ defined?
Or say 'In addition horses examined during the pilot study which fulfilled the inclusion criteria were also included.'
We changed it to this. Thank you.
Line 82 Twenty (numbers should be written in full at the beginning of a sentence)
This was changed.
Line 82 How can you say with certainty that the lameness was unilateral unless lameness was abolished by diagnostic anaesthesia and did not switch to the contralateral limb?
If this was not done then this should be mentioned as a limitation of your study.
It was done and mentioned in the text already.
You imply in line 96 that diagnostic anaesthesia was used. This would be better mentioned here.
e.g., All lame horses underwent diagnostic anaesthesia of the lame lime and in none did lameness switch to the contralateral limb.
This was moved here and the sentence was shortened. Thank you.
Line 90 11 is a cattle reference not a horse reference! I don't think this is particularly appropriate.
Unfortunately we can’t find reference 11 in Line 90, but in general we think, if the topic, which we are doing in horses, is already mentioned in other species, this is a good thing to mention.
This sentence does not read easily - please rephrase.
This was rephrased to: Lameness was graded using the American Association of Equine Practitioners Lameness Scale (AAEP) [16,17] (Appendix 1: The American Association of Equine Practitioners lameness grading system). Half units were described when there was more than one evaluating veterinarian [1] and were allowed in this study due to the fact that two veterinarians assessed the lameness. The score was averaged if different.
All horses were assessed in walk and trot in a straigt line on a firm ground. In addition, they were lunged at walk, trot and canter on a soft surface, if their lameness allowed it (canter was not always possible for those with grade 4 lameness).
Line 95 when (not where)
This was changed.
Lines 96-98 - see previous comment
Ok.
Line 102 AAEP scale
This was changed.
I think that you need to explain in the figure legend what is meant by, for example, 2-3/5, because this is not consistent with what you have said in the test when you said you used halves and used the mean values for the two observers. So in this Table if you say 2-3 do you mean that one observer graded the lameness as 2 and the other as 3? There is lack of consistency in presentation in Tables 1 & 2.
The tables were changed to be more consistent.
Line 104 approximately
This was changed.
Line 116 The method would be better than the way
This was changed.
Line 149 were performed
This was changed.
Line 152 minutes (not minuets!)
This was changed.
Line 159 were analysed
This was changed.
Line 166 were calculated
This was changed.
Line 172 are summarised
This was changed.
In the legend for both Tables 1 & 2 the term 'Resting relation' needs to be defined. Remember the legend must be able to be read independently from the text
The definition of resting relation was added to the legends. Also we made them more consistent.
Table 2 we need to know what is meant by the asymmetry measurement given of the Lameness Locator - was this MinDiff, MaxDiff?
Yes. And it was added to the text as an definition: Asymmetry means the minimal and maximal difference of the Equinosis Lameness Locator in millimeter.
It is rather confusing that the Vet Score is different to that in Table 1 because Table 1 gives a range whereas as Table 2 gives halves
This was changed to be more consistent.
Line 183 This needs to be expressed differently
The lame limb was loaded for less time than the non-lame limb (p=0.035).
Do you mean different because of the expressions of lame and non-lame limb? We thought we have to be consistent in the terminology with these words.
Line 185-6'The total of the lame horses (13 ± 185 11%) lied down significantly longer (p= 0.01) than the group of non-lame horses (3 ± 6%) ' Please rephrase this - I don't know what you are saying here
Was rephrased: “The lame horses (13 ± 11%) lied down significantly longer (p=0.01) than the non-lame horses (3 ± 6%).”
Line 187 - 9 ‘Resting relation was defined as non-lame resting time / lame resting time within the lame group of horses and right limb resting time / left limb resting time within the non-lame group of horses. ‘
This repeats the M&M and doesn't belong here.
This was removed here.
Figure 2 The large SD and the overlap between the two groups of horses needs to be discussed.
The limitations of this study were pointed out in the discussion.
Legend for Fig. 2 Relative resting time needs to be defined in the legend
This is now defined in the legend.
I think that you misunderstood my comments about the outlier. The reason for this outlier needs to be included in the Discussion. The number of the horse in the figure should marry up with the horse number in the tables - it is otherwise confusing even if explained in the legend - change the figure if this is actually horse 6
The figures have been changed. Hopefully it is less confusing now. Please also see the outliers being discussed in the discussion.
Line 195 All Results should be written in the past tense
This was changed.
Line 195 what is meant by regularly in this context?
Regularly means that the resting "events" took place in the same horse in the same way. No results present this. It was deleted.
Lines 196 and 199 - you don't need to refer back to the hypotheses in the results - you should merely state the results
This was deleted wherever it was mentioned.
Figure 3 Please amend your figure so that the outliers are the correct horse numbers - do not create unnecessary confusion - even if you attempt to explain this in the legend. Keep things consistent and simple.
The numbers in the figures have been changed.
Line 209'(5040 ± 4268) 209 compared to the non-lame horses (996 ± 2108). What units are these numbers?
These units are still the seconds and our mistake to not changing them here to minutes as well. Sorry.
Line 214 The asymmetry did not show;
the mean asymmetry was different between the lame and non-lame horses.
This was rephrased to: The mean asymmetry evaluated by the lameness locator (ALL) was different (p= 0.025) between the lame (10.5 ± 5.6 mm) and non-lame horses (5.8 ± 3.4 mm) (Figure 4).
Here and elsewhere it is unnecessary to say significant or significantly if the p value is <0.05.
These words have been removed.
Figure 4 20 horses with unilateral hindlimb lameness and 10 non-lame horses
This was changed.
This Discussion remains FAR TOO LONG - it must be shortened and made much more focussed and concise as indicated previously
Please see the new discussion.
You need to justify how you can call non-lame horses which have asymmetry values outside the so-called normal range & sometimes higher than those observed in the lame horses
Please see added: The horses were included/separated based on the Vet Score. A perfect match between Vet Score and ALL is very unlikely. However, the statistical comparison showed a significant correlation between Vet Score and ALL.
You need to think about why the so-called non-lame horses were lying down less than the lame horses - this is abnormal - & needs to be thought of like that
Measurements for this study were taken at night to capture natural resting patterns with minimal disturbance, although the clinic setting may have influenced the results due to the unfamiliar environment. Our results may reflect that the horses are in an unfamiliar environment, especially the group of non-lame horses that do not lie down. Uncertainty in the data can certainly be reduced if the horses are analyzed in their usual environment.
Lines 224 -7 It is unnecessary to repeat the hypotheses
You could say
In accordance with our hypotheses non-lame horses rested their hindlimbs equally etc.
This was removed and changed to your comment. Thank you.
Line 227 I think that you have to consider the duration of time for which the lame horses had been lame - how might that influence your results?
Horses might get used to the pain. So, if the lameness goes on for a longer time, they might show less pain and maybe even a lower grade. Which doesn't matter for this study, because we just take in account, if they are lame or not.
Line 228 - 242 This is all conjecture and should be expressed much more concisely.
Please see the changed text/discussion.
Line 247 'How this influences the actual resting patterns would be a good investment for future studies. ' I don't think investment is an appropriate word in this context
The word investment was removed.
Line 250-1 However, this would also be a consideration that can be taken into account in future experiments on this topic.
You did not do an experiment! I think that you mean that these factors merit further investigation
Yes. This was changed and the word experiment is not used anymore.
Line 252 - 263 - much of this is irrelevant and /or speculation
Please see the changed text.
All you need to say that variation among breeds in your study may have influenced the results
Please see the changed text.
Reference 30 is a review article & I do not believe that it provides any evidence about specific breed differences
This was removed.
Line 259 I don't think that reference 32 is applicable here
This was removed.
Lines 263-267 This has to be put into the context that we know that chronic orthopaedic pain can be a reason for failure to lie down, sleep deprivation and a tendency to collapse
Please see the changed text/discussion.
Line 270 smaller than what?
Should say small.
Line 276 'Whether the time of signaling in horses increases with a higher degree of lameness would have to be specifically investigated in a new experiment.’ What does this mean?
Please see the changed text.
Line 284 -88 This could be expressed much more concisely
Please see the changed text.
Lines 289 - 295 This is not really relevant - what you need to discuss is the overlap between measured asymmetry between the lame horses and the non-lame horses and the relationship between what you measured and the published threshold values for lameness
This was deleted. And more precise discussion was added.
Line 307 led , not lead
This was changed.
Lines 310 - 14 This is not really necessary
This was removed.
Line 332 - 336 this belongs earlier
This was moved earlier.
Line 336 - As I have said before I think it is more remarkable that your non-lame horses were so abnormal in their lying down patterns. Why?
Please see the changed text. And the new parts in the discussion.
Lines 344 - 356 Huge amount of repetition here
Please see the changed text.
This Discussion remains unfocussed and there are long rambling sections that are not related directly to the results of your study - you must relate your comments back to your results
Please see the new discussion.
Avoid repeating what you've already said in the discussion in the conclusions
Please see the new conclusions.
Comments on the Quality of English Language
Still not good enough, although much improved.
The language was checked by a native speaker again.
Submission Date
03 June 2024
Date of this review
08 Sep 2024 05:10:44